# What Linear Probes Miss: Multi-View Probing for Weight-Space Learning

**Eunwoo Heo** [* 1]   **Kyeongkook Seo** [* 1]   **Jaejun Yoo** [1]

## Abstract

The explosive growth of open-source model repositories has created a Model Jungle, where checkpoints are frequently shared without adequate documentation or metadata. While weight-space learning offers a pathway to identify and analyze these models directly from their parameters, processing full-scale weights is computationally prohibitive. Probing-based methods have emerged as a lightweight alternative, extracting permutation-equivariant representations via learnable probe vectors. However, existing probing methods are limited by a single-view design: they capture first-order structures but fail to encode the rich, higher-order correlation patterns inherent in row–column interactions. To bridge this gap, we introduce MVProbe, a multi-perspective probing framework that synthesizes first-order signals with interaction-aware (Gram-based) views. Our approach is theoretically grounded; we analyze the scaling laws of different probing orders to derive a principled standardization and fusion strategy that ensures balanced contributions from all branches. On the Model Jungle benchmark, MVProbe[1] consistently outperforms the state-of-the-art ProbeX across diverse architectures, including discriminative backbones (ResNet, SupViT, MAE, DINO) and large-scale generative LoRA adapters (Stable Diffusion LoRA).

## 1. Introduction

With the rapid advancement of deep learning, users increasingly share task-specific fine-tuned model checkpoints on public platforms such as Hugging Face and CivitAI.

This ecosystem enables users to leverage specialized models without incurring the high costs of large-scale training. However, identifying suitable models among millions of available checkpoints remains challenging, particularly when model documentation is incomplete or missing—approximately 23% of models on Hugging Face (Pepe et al., 2024). In such cases, key properties—including training data distribution or generalization capability—are not directly accessible, and assessing them typically requires costly evaluation, making reliable model selection difficult.

To address this challenge, *weight-space learning* has emerged as a paradigm designed to infer model properties directly from parameters, independent of external metadata (Eilertsen et al., 2020; Unterthiner et al., 2020). However, direct learning from raw weights faces two key challenges: (i) processing flattened model weights is computationally burdensome, and (ii) extracting information from high-dimensional weight spaces is nontrivial. To mitigate these issues, probing-based approaches analyze model behavior by injecting learnable probe vectors, rather than operating directly on the model weights (Herrmann et al., 2024; Kofinas et al., 2024; Kahana et al., 2025; Horwitz et al., 2025). While early methods showed promise on small-scale benchmarks (e.g., MNIST, CIFAR-10), they often struggle to scale (Zhang et al., 2023; Navon et al., 2023; Kofinas et al., 2024; Lim et al., 2024). Recently, ProbeX (Horwitz et al., 2025) introduced single-layer probing, achieving scalability to large models by analyzing selected weight matrices.

Despite its efficiency, single-layer probing in existing methods relies on a single first-order projection along one probing direction, which can be insufficient to characterize weight matrices. In particular, first-order projections exhibit inherent ambiguity: distinct weight matrices may induce indistinguishable probe responses, causing them to collapse to the same representation. Moreover, a single linear view fails to capture higher-order structure, such as correlations expressed through row–column pairwise interactions.

In this work, we show that distinct weight matrices can yield identical representations under first-order probing, rendering them indistinguishable (Theorem 4.1). To overcome this limitation while retaining scalability, we introduce Multi-View Probe (MVProbe), a *multi-perspective* framework that augments first-order views with interaction-aware (Gram-

---
[*]Equal contribution   [1]Graduate School of Artificial Intelligence, Ulsan National Institute of Science and Technology, Ulsan, Republic of Korea. Correspondence to: Jaejun Yoo <jaejun.yoo@unist.ac.kr>.

*Proceedings of the 43$^{rd}$ International Conference on Machine Learning*, Seoul, South Korea. PMLR 306, 2026. Copyright 2026 by the author(s).

[1]Code is available at https://github.com/AI-hew-math/MVProbe.

based) views. Concretely, MVProbe extracts complementary features from both row and column perspectives at first and second order. To ensure balanced fusion across views with differing intrinsic scales, we derive a principled per-sample standardization scheme grounded in our scaling analysis (Theorem 4.3).

Our approach is motivated by two key insights. First, from the *kernel methods* perspective, the Gram matrix encodes pairwise similarity structure that first-order projections cannot capture. By probing through these kernel matrices, we access rich correlation information about how rows (or columns) relate to each other. Second, from the *landmark points* perspective in manifold learning (Silva & Tenenbaum, 2003; De Silva & Tenenbaum, 2004), our probe vectors serve as learnable reference directions that define a coordinate system for the weight matrix. With four probing branches (row/column direct projections and their Gram-based counterparts), MVProbe provides complementary views of the induced coordinate system, yielding a comprehensive characterization of weight matrix geometry.

We focus on the single-layer probing regime introduced by ProbeX, where each model is represented using the weight matrix from a representative layer. We validate MVProbe on the Model Jungle dataset [2], covering modern architectures such as ResNet-101 and ViT-Base, as well as Stable Diffusion LoRA checkpoints. MVProbe surpasses existing baselines by a significant margin across various architectures. Notably, MVProbe demonstrates robust performance across all layers where ProbeX fails to learn informative representations, producing near-random performance. Furthermore, the Stable Diffusion LoRA experiments show that MVProbe extends beyond conventional vision-backbone checkpoints to generative-model LoRA checkpoints with substantially larger label sets.

We summarize our contributions as follows:

- We characterize a fundamental limitation of first-order single-view probing, showing that distinct weight matrices can become indistinguishable under the same probe representation.

- We propose MVProbe, a multi-perspective probing framework that combines first-order projections with Gram-based interaction views, together with a principled per-sample standardization scheme for balanced fusion across scales.

- MVProbe achieves state-of-the-art performance on Model Jungle and exhibits robust layer-wise behavior, including layers that are weakly informative for probing.

---

[2]https://huggingface.co/ProbeX

## 2. Related Work

**Weight-space learning.** Early works have focused on understanding neural networks and learning representations by predicting model properties, such as training dataset or generalization error using flattened parameters or weight statistics (Eilertsen et al., 2020; Unterthiner et al., 2020; Schürholt et al., 2021). However, these methods often destroy critical structural relationships and ignore the inherent permutation equivariance of neural networks, where reordering hidden neurons yields a functionally equivalent network (Navon et al., 2023; Zhou et al., 2023; Zhao et al., 2025). To address these challenges, probing-based methods have emerged as a dominant paradigm. By characterizing model weights through their input-output behavior, these methods achieve permutation equivariance naturally. Probe-Gen (Kahana et al., 2025) introduces a probe generator to ensure probe vectors have a shared latent space. Neural Graph (Kofinas et al., 2024) treats model weights as graphs to leverage graph neural networks while maintaining neural symmetry and structural relationships. Most recently, ProbeX (Horwitz et al., 2025) explores a large-scale benchmark of fine-tuned models—ResNet101 (He et al., 2016), Supervised ViT (Dosovitskiy et al., 2021), Masked Autoencoder (He et al., 2022), and DINO (Caron et al., 2021)—for predicting the fine-tuning label set from model weights.

Weight-space learning has also expanded into diverse tasks, including LoRA weight classification (Putterman et al., 2024), model derivation recovery (Horwitz et al., 2024), weight generation (Zeng et al., 2026; Schürholt et al., 2022), and model merging (Wortsman et al., 2022; Yadav et al., 2023). While these tasks demonstrate the breadth of the field, they still face the fundamental limitation in capturing sufficient geometric information from weights. Our work addresses this gap by introducing a multi-perspective probing framework that integrates first-order (row/column projections) and second-order (kernel) views, providing a more comprehensive representation than prior first-order-only approaches.

**Intrinsic representations.** Kernel methods (Schölkopf & Smola, 2002) are classic techniques that represent data through Gram matrices, which encode pairwise similarity structure between samples. By implicitly mapping data into high-dimensional feature spaces, the kernel trick allows linear algorithms to model complex non-linear structures. This representation has been widely used in machine learning, as it captures rich relational information through inner products in the induced feature space.

Closely related to our work are landmark-based representations, which approximate global data geometry using a small set of reference points commonly referred to as *landmark points* (Silva & Tenenbaum, 2003; De Silva & Tenenbaum, 2004). Landmark-based methods approximate pairwise rela-

tionships by measuring distances to a subset of representative points, enabling scalable geometric representations of high-dimensional data. These approaches share conceptual similarities with kernel methods, as both rely on relational information rather than explicit feature coordinates. In this paper, probe responses could be interpreted as landmark points of given weight matrices that serve as anchors for well-defined representation vectors. Recent work on gradient processing further suggests that second-order geometric information—e.g., curvature captured by the Fisher Information Matrix—encodes signals inaccessible to first-order methods (Gelberg et al., 2026), motivating curvature-aware weight-space representations.

# 3. Background

## 3.1. Problem Definition

We consider the task of weight-space learning, where the goal is to predict properties of neural networks directly from their weights. Formally, we have a dataset $\mathcal{D} = \{(\mathbf{W}^{(i)}, \mathbf{y}^{(i)})\}_{i=1}^{N}$, where $\mathbf{W}^{(i)}$ denotes the weights of the $i$-th neural network and $\mathbf{y}^{(i)} \in \{0,1\}^{C}$ denotes its attribute vector, where $C$ is the number of target classes. We consider a single layer and denote its weight matrix by $\mathbf{X} \in \mathbb{R}^{m \times n}$, where $m$ and $n$ are the numbers of output and input neurons, respectively. $\mathbf{X}$ is taken to be the most informative layer, i.e., the single layer that yields the best classification performance when used alone. Our goal is to predict the attribute vector $\mathbf{y}$ solely from $\mathbf{X}$.

## 3.2. Probing for Weight-Space Learning

A key challenge in weight-space learning is the permutation equivariance of neural networks: permuting neurons in hidden layers changes the weight representation but preserves the network function (Navon et al., 2023). Probing-based methods (Herrmann et al., 2024; Kofinas et al., 2024; Kahana et al., 2025; Horwitz et al., 2025) address this by characterizing networks through their input-output behavior, which is inherently permutation-equivariant (Herrmann et al., 2024; Kahana et al., 2025).

Specifically, we learn a set of $r$ probe vectors $\mathbf{u}_1, \ldots, \mathbf{u}_r \in \mathbb{R}^n$, collected as columns of a probe matrix $\mathbf{U} \in \mathbb{R}^{n \times r}$. These are passed through the weight matrix to obtain probe responses:

$$\mathbf{S} = \mathbf{X}\mathbf{U} \in \mathbb{R}^{m \times r}. \quad (1)$$

Each column $\mathbf{S}_{:,k} = \mathbf{X}\mathbf{u}_k \in \mathbb{R}^m$ captures the responses of the output neurons to the $k$-th probe vector. These responses are then processed by an encoder $\phi$ to predict model attributes:

$$\hat{\mathbf{y}} = \phi(\mathbf{S}). \quad (2)$$

The probe vectors and encoder are jointly trained to minimize a task-specific loss.

## 3.3. Limitations of First-Order Probing

While probing yields representations that are equivariant to neuron permutations, the standard formulation $\mathbf{S} = \mathbf{X}\mathbf{U}$ has two fundamental limitations: (1) Row–Column asymmetry and (2) Missing pairwise interaction structure. Note that row–column asymmetry below is distinct from permutation equivariance: while permutation equivariance refers to symmetry under independent permutations of input and output neurons, the row–column asymmetry is not a model symmetry but a property of which axis the probe interrogates.

**Row–Column asymmetry.** Standard probing computes first-order responses $\mathbf{S} = \mathbf{X}\mathbf{U} \in \mathbb{R}^{m \times r}$, which aggregates each output neuron's incoming weights along a small number of probe directions. This yields a row-centric sketch: it captures how output neurons aggregate inputs along the probe directions, but it does not directly characterize the column-wise perspective—namely, the geometry on the input side and how input coordinates are coupled through the layer. Equivalently, if we interpret $\mathbf{X}$ as an adjacency matrix between input and output neurons, $\mathbf{X}\mathbf{U}$ summarizes row-side aggregation patterns under the chosen probes, while variations on the column side that do not affect these aggregates remain unobserved. Consequently, probing can be blind to components of $\mathbf{X}$ that fall in the nullspace of $\mathbf{U}$: for any $\mathbf{Z}$ satisfying $\mathbf{Z}\mathbf{U} = 0$, we have $(\mathbf{X} + \mathbf{Z})\mathbf{U} = \mathbf{X}\mathbf{U}$, so the probe responses cannot distinguish $\mathbf{Z}$ from $\mathbf{X} + \mathbf{Z}$.

**Missing pairwise interaction structure.** First-order responses capture only direct interactions between $\mathbf{X}$ and individual probe directions. They do not explicitly capture pairwise relationships among output neurons or among input coordinates, which is crucial for diagnosing redundancy and collective modes—e.g., whether groups of neurons share similar input-weighting patterns or whether input coordinates are strongly coupled through the layer. Such behavior is primarily governed by the induced similarity structure (e.g., Gram matrices) and its spectral geometry (e.g., effective rank and conditioning), beyond what isolated directional responses can capture.

These limitations suggest that an effective weight-space representation should (i) treat rows and columns symmetrically, and (ii) explicitly encode pairwise similarities among neurons/features.

# 4. Method

We propose Multi-View Probe (MVProbe), a multi-perspective probing framework that extracts complementary features from weight matrices via four branches. By combining first-order probing (direct row/column projections) with second-order probing through Gram operators, we obtain a richer representation that captures both local geometry

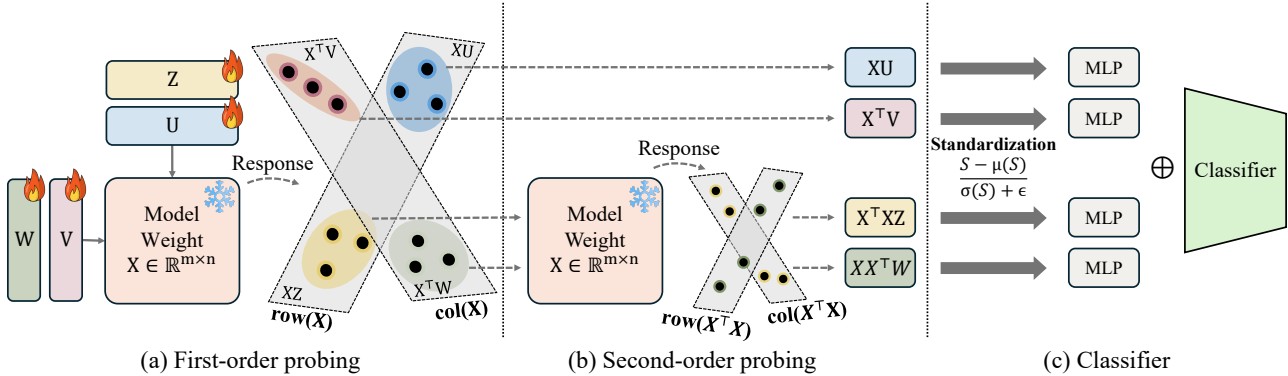

(a) First-order probing      (b) Second-order probing      (c) Classifier

*Figure 1.* **Overview of MVProbe.** Given a weight matrix $\mathbf{X} \in \mathbb{R}^{m \times n}$, MVProbe extracts probe responses from four complementary views. (a) *First-order probing:* learnable probes $\mathbf{U}$ and $\mathbf{V}$ produce row- and column-space responses $\mathbf{XU}$ and $\mathbf{X}^\top \mathbf{V}$. We also compute $\mathbf{XZ}$ and $\mathbf{X}^\top \mathbf{W}$ as intermediate projections for the second-order branches. (b) *Second-order probing:* applying $\mathbf{X}^\top$ and $\mathbf{X}$ once more yields Gram-based responses $\mathbf{X}^\top \mathbf{XZ}$ and $\mathbf{XX}^\top \mathbf{W}$, capturing column- and row-wise correlation structure. (c) All responses are standardized, processed by branch MLPs, concatenated ($\oplus$), and fed into a shared encoder followed by a classifier head.

and global correlation patterns. A schematic overview of MVProbe is shown in Figure 1.

### 4.1. Four-Branch Probing Architecture

We propose four complementary probing branches that capture different structural aspects of the weight matrix. Each branch uses its own learnable probe matrix and projection layer. A schematic illustration of these structural aspects is provided in Appendix F.

#### 4.1.1. FIRST-ORDER BRANCHES

First-order branches directly probe the row and column structure of $\mathbf{X} \in \mathbb{R}^{m \times n}$.

**Row probing ($\mathbf{XU}$).** With probe matrix $\mathbf{U} \in \mathbb{R}^{n \times r}$, we compute:

$$\mathbf{S}^{\text{row}} = \mathbf{XU} \in \mathbb{R}^{m \times r}. \tag{3}$$

Each row $\mathbf{S}^{\text{row}}_{i,:}$ is an $r$-dimensional summary of the probe responses of the $i$-th output neuron. This branch captures how each output neuron aggregates inputs along the probe vectors.

**Column probing ($\mathbf{X}^\top \mathbf{V}$).** With a probe matrix $\mathbf{V} \in \mathbb{R}^{m \times r}$, we compute:

$$\mathbf{S}^{\text{col}} = \mathbf{X}^\top \mathbf{V} \in \mathbb{R}^{n \times r}. \tag{4}$$

Each row $\mathbf{S}^{\text{col}}_{j,:}$ summarizes how the $j$-th input neuron connects to output neurons along the $r$ probe vectors.

#### 4.1.2. SECOND-ORDER (KERNEL) BRANCHES

While first-order branches capture direct projections of probe vectors through $\mathbf{X}$, they may miss higher-order correlations between rows or columns. A natural way to capture such information is through Gram matrices derived from $\mathbf{X}$. We introduce second-order branches that probe through

*kernel matrices* (Gram matrices), which encode pairwise similarity structure. A formal connection to kernel methods is given in Appendix C.

**Row kernel probing ($\mathbf{XX}^\top \mathbf{W}$).** Let $\mathbf{W} \in \mathbb{R}^{m \times r}$ be a probe matrix. With the row Gram matrix $\mathbf{K}_{\text{row}} = \mathbf{XX}^\top \in \mathbb{R}^{m \times m}$, we define the row-kernel response as

$$\mathbf{S}^{\text{row-kernel}} = \mathbf{K}_{\text{row}} \mathbf{W} \in \mathbb{R}^{m \times r}. \tag{5}$$

The $k$-th probe column $\mathbf{w}_k$ of $\mathbf{W}$ assigns coefficients over output neurons (rows) of $\mathbf{X}$. Its response is the $k$-th column of $\mathbf{S}^{\text{row-kernel}}$, i.e., $\mathbf{S}^{\text{row-kernel}}_{:,k} = \mathbf{K}_{\text{row}} \mathbf{w}_k \in \mathbb{R}^m$, whose $i$-th entry is

$$(\mathbf{K}_{\text{row}} \mathbf{w}_k)_i = \sum_{j=1}^{m} w_{j,k} \langle \mathbf{X}_{i,:}, \mathbf{X}_{j,:} \rangle. \tag{6}$$

Here, $\langle \cdot, \cdot \rangle$ denotes the standard Euclidean inner product. Equivalently,

$$\mathbf{K}_{\text{row}} \mathbf{w}_k = \sum_{j=1}^{m} w_{j,k} (\mathbf{K}_{\text{row}})_{:,j}, \tag{7}$$

so $\mathbf{w}_k$ specifies a weighted combination of the *similarity-to-row profiles* (the columns of $\mathbf{K}_{\text{row}}$). Thus, for each neuron $i$, $(\mathbf{K}_{\text{row}} \mathbf{w}_k)_i$ aggregates its pairwise similarities to all output neurons.

**Column kernel probing ($\mathbf{X}^\top \mathbf{XZ}$).** The column Gram matrix $\mathbf{K}_{\text{col}} = \mathbf{X}^\top \mathbf{X} \in \mathbb{R}^{n \times n}$ encodes pairwise similarities between input features (columns) of $\mathbf{X}$. Let $\mathbf{Z} \in \mathbb{R}^{n \times r}$ be a probe matrix and $\mathbf{z}_k$ its $k$-th column. The column-kernel response is

$$\mathbf{S}^{\text{col-kernel}} = \mathbf{K}_{\text{col}} \mathbf{Z} \in \mathbb{R}^{n \times r}, \tag{8}$$

whose $p$-th entry for probe $k$ is

$$(\mathbf{K}_{\mathrm{col}}\mathbf{z}_k)_p = \sum_{q=1}^{n} z_{q,k} \langle \mathbf{X}_{:,p}, \mathbf{X}_{:,q} \rangle. \tag{9}$$

Equivalently,

$$\mathbf{K}_{\mathrm{col}}\mathbf{z}_k = \sum_{q=1}^{n} z_{q,k} (\mathbf{K}_{\mathrm{col}})_{:,q}, \tag{10}$$

so $\mathbf{z}_k$ forms a weighted combination of *similarity-to-column profiles* (the columns of $\mathbf{K}_{\mathrm{col}}$). Thus, for each input feature $p$, $(\mathbf{K}_{\mathrm{col}}\mathbf{z}_k)_p$ aggregates its pairwise similarities to input features.

Compared to first-order probing (e.g., $\mathbf{X}\mathbf{U}$ or $\mathbf{X}^\top\mathbf{V}$), the kernel branches $\mathbf{S}^{\mathrm{row\text{-}kernel}} = \mathbf{K}_{\mathrm{row}}\mathbf{W}$ and $\mathbf{S}^{\mathrm{col\text{-}kernel}} = \mathbf{K}_{\mathrm{col}}\mathbf{Z}$ explicitly encode second-order correlation structure among output neurons (rows) and among input features (columns), respectively.

## 4.2. Geometric Interpretation: Probes as Landmarks

We provide a geometric interpretation of our four-branch architecture through the lens of *landmark-based representations* (Silva & Tenenbaum, 2003; De Silva & Tenenbaum, 2004). In manifold learning, landmark points serve as reference locations that define a coordinate system for high-dimensional data.

In our framework, probe vectors play an analogous role:

- **First-order probes** define landmark *directions* in the ambient space. The probe response $\mathbf{X}\mathbf{U}$ measures how each row of $\mathbf{X}$ projects onto these directions.

- **Second-order probes** define landmark *combinations* in the kernel space. The response $\mathbf{X}\mathbf{X}^\top\mathbf{W}$ measures how each row relates to other rows through the lens of these landmark combinations.

By learning these landmarks end-to-end for the classification task, the model discovers reference points that are most informative for distinguishing between different weight matrix configurations.

## 4.3. Architecture Details

**Per-sample standardization.** We standardize each probe response matrix per sample:

$$\tilde{\mathbf{S}} = \frac{\mathbf{S} - \mu(\mathbf{S})}{\sigma(\mathbf{S}) + \epsilon}, \tag{11}$$

where $\mu(\mathbf{S})$ and $\sigma(\mathbf{S})$ are the mean and standard deviation computed over all elements of $\mathbf{S}$. This operation mitigates

within-sample scale imbalance across branches and also stabilizes across-sample variations caused by differing weight magnitudes.

**Branch-specific projection.** Each branch has its own projection layer $\pi_i$ that maps the probe responses to a common dimension $d$:

$$\mathbf{f}_i = \mathrm{ReLU}(\pi_i(\tilde{\mathbf{S}}^{(i)})) \in \mathbb{R}^d, \tag{12}$$

where $\tilde{\mathbf{S}}^{(i)}$ is flattened before projection.

**Shared encoder and classifier.** Let $\mathbf{f} = [\mathbf{f}_1; \mathbf{f}_2; \mathbf{f}_3; \mathbf{f}_4] \in \mathbb{R}^{4d}$ denote the concatenation of the four branch features. We first map $\mathbf{f}$ to a shared latent representation using a shared encoder $\psi : \mathbb{R}^{4d} \to \mathbb{R}^{d_h}$, and then produce class logits using a classifier head $\phi : \mathbb{R}^{d_h} \to \mathbb{R}^C$:

$$\hat{\mathbf{y}} = \phi(\psi(\mathbf{f})) \in \mathbb{R}^C, \tag{13}$$

where $d_h$ is the hidden dimension and $C$ is the number of classes.

## 4.4. Training Objective

We train the model with a standard multi-label classification loss:

$$\mathcal{L} = \mathcal{L}_{\mathrm{BCE}}(\hat{\mathbf{y}}, \mathbf{y}), \tag{14}$$

where $\mathcal{L}_{\mathrm{BCE}}$ is the binary cross-entropy loss and $\mathbf{y}$ is the ground-truth label vector.

## 4.5. Theoretical Analysis

We provide theoretical justifications for two central components of our approach: (1) necessity of second-order branches, and (2) importance of per-sample standardization.

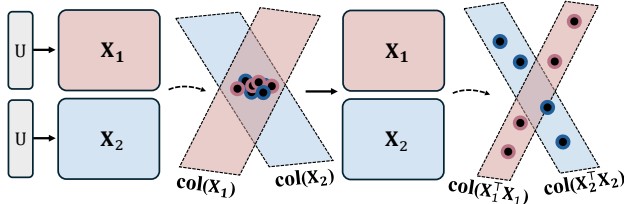

*Figure 2.* **Conceptual illustration of Theorem 4.1.** With the same probe matrix $\mathbf{U}$, two distinct weight matrices $\mathbf{X}_1$ and $\mathbf{X}_2$ may produce first-order probe responses that are indistinguishable under standard probing. Theorem 4.1 shows that second-order, Gram-based representations can separate such cases.

**Theorem 4.1** (Expressiveness of Second-Order Probes). *Let* $\mathbf{U} \in \mathbb{R}^{n \times r}$ *be a probe matrix with* $\mathrm{rank}(\mathbf{U}) = r < n$. *Define the first-order and second-order features:*

$$\Phi_1(\mathbf{X}) := \mathbf{X}\mathbf{U} \in \mathbb{R}^{m \times r}, \quad \Phi_2(\mathbf{X}) := (\mathbf{X}^\top\mathbf{X})\mathbf{U} \in \mathbb{R}^{n \times r}. \tag{15}$$

*Then there exist distinct matrices* $\mathbf{X}_1 \neq \mathbf{X}_2$ *such that*

$$\Phi_1(\mathbf{X}_1) = \Phi_1(\mathbf{X}_2) \quad but \quad \Phi_2(\mathbf{X}_1) \neq \Phi_2(\mathbf{X}_2). \quad (16)$$

*Consequently, when $r < n$, adding second-order branches can separate weight matrices that are indistinguishable to first-order probing alone (depicted in Figure 2).*

*Proof sketch.* Since $\mathrm{rank}(\mathbf{U}) = r < n$, there exists $\mathbf{w} \neq \mathbf{0}$ such that $\mathbf{w}^\top \mathbf{U} = \mathbf{0}^\top$. Construct $\mathbf{X}_1 = \mathbf{a}\mathbf{v}^\top$ and $\mathbf{X}_2 = \mathbf{a}(\mathbf{v} + \mathbf{w})^\top$ where $\mathbf{v}^\top \mathbf{U} \neq \mathbf{0}$. Then $\mathbf{X}_1 \mathbf{U} = \mathbf{X}_2 \mathbf{U}$, but $(\mathbf{X}_2^\top \mathbf{X}_2)\mathbf{U} - (\mathbf{X}_1^\top \mathbf{X}_1)\mathbf{U} = \|\mathbf{a}\|_2^2 \mathbf{w}\mathbf{v}^\top \mathbf{U} \neq \mathbf{0}$. Full proof in Appendix B. □

Empirical evidence for this theorem—embedding-overlap and neighborhood-recovery analyses on failure layers—is provided in Appendix H.

Symmetrically, the column-side probe $\mathbf{X}^\top \mathbf{V}$ is also non-redundant with $\mathbf{X}\mathbf{U}$:

**Theorem 4.2** (Transpose-Complement Non-Redundancy). *Let $\mathbf{U} \in \mathbb{R}^{n \times r}$ and $\mathbf{V} \in \mathbb{R}^{m \times r}$ be probe matrices with $\mathrm{rank}(\mathbf{U}) = r < n$. Then there exist distinct $\mathbf{X}_1, \mathbf{X}_2 \in \mathbb{R}^{m \times n}$ such that*

$$\mathbf{X}_1 \mathbf{U} = \mathbf{X}_2 \mathbf{U} \quad but \quad \mathbf{X}_1^\top \mathbf{V} \neq \mathbf{X}_2^\top \mathbf{V}. \quad (17)$$

*Proof sketch.* Pick $\mathbf{w} \in \mathbb{R}^n$ with $\mathbf{w}^\top \mathbf{U} = \mathbf{0}^\top$ (exists by $\mathrm{rank}(\mathbf{U}) < n$) and $\mathbf{a} \in \mathbb{R}^m$ with $\mathbf{V}^\top \mathbf{a} \neq \mathbf{0}$. Set $\mathbf{X}_2 = \mathbf{X}_1 + \mathbf{a}\mathbf{w}^\top$. Then $\mathbf{X}_2 \mathbf{U} = \mathbf{X}_1 \mathbf{U} + \mathbf{a}(\mathbf{w}^\top \mathbf{U}) = \mathbf{X}_1 \mathbf{U}$, while $\mathbf{X}_2^\top \mathbf{V} - \mathbf{X}_1^\top \mathbf{V} = \mathbf{w}(\mathbf{a}^\top \mathbf{V}) \neq \mathbf{0}$. Full proof in Appendix B. □

While second-order branches provably resolve such ambiguities, naively processing multi-order responses introduces a scale imbalance. The following theorem quantifies this discrepancy, motivating our per-sample standardization.

**Theorem 4.3** (Scale Imbalance in Multi-Order Branches). *Let $\mathbf{X} \in \mathbb{R}^{m \times n}$ have i.i.d. entries $X_{ij} \sim \mathcal{N}(0, \sigma^2)$. For first-order response $\mathbf{S}^{(1)} = \mathbf{X}\mathbf{U}$ and second-order response $\mathbf{S}^{(2)} = (\mathbf{X}\mathbf{X}^\top)\mathbf{W}$ with unit-norm probe columns, the expected scale ratio is:*

$$\frac{\mathbb{E}[\|\mathbf{S}^{(2)}\|_F^2]}{\mathbb{E}[\|\mathbf{S}^{(1)}\|_F^2]} = \frac{n(n+m+1)}{m}\sigma^2 = O(n\sigma^2), \quad (18)$$

*where $\|\cdot\|_F$ denotes the Frobenius norm. In particular, when $n\sigma^2 \gg 1$, unstandardized concatenation $[\mathbf{S}^{(1)}, \mathbf{S}^{(2)}]$ is dominated by the second-order branch.*

*Proof sketch.* Direct computation shows $\mathbb{E}[\|\mathbf{S}^{(1)}\|_F^2] = mr\sigma^2$ and $\mathbb{E}[\|\mathbf{S}^{(2)}\|_F^2] = rn(n+m+1)\sigma^4$. The ratio follows immediately. Full proof in Appendix B. □

**Corollary 4.4** (Per-sample Standardization Equalizes Branch Contributions). *Per-sample standardization $\tilde{\mathbf{S}} =$*

$(\mathbf{S} - \mu_\mathbf{S})/\sigma_\mathbf{S}$ *yields $\|\tilde{\mathbf{S}}\|_F^2$ equal to the number of elements of $\mathbf{S}$, independent of the probing order. Consequently, applying this standardization independently to each branch (e.g., $\tilde{\mathbf{S}}^{(1)}$ and $\tilde{\mathbf{S}}^{(2)}$) mitigates the scale imbalance in Theorem 4.3 and prevents higher-order branches from dominating the concatenated representation.*

Theorem 4.3 motivates standardization: without it, naive concatenation $[\mathbf{S}^{(1)}, \mathbf{S}^{(2)}]$ can be skewed toward second-order responses; empirically, Table 3 shows that standardization consistently improves accuracy.

## 5. Experiments

**Datasets.** We evaluate MVProbe primarily on the Model Jungle dataset [3] (Horwitz et al., 2025), which contains fine-tuned weights from four foundation model architectures: ResNet101 (He et al., 2016), SupViT (Dosovitskiy et al., 2021), MAE (He et al., 2022), and DINO (Caron et al., 2021). Each model is fine-tuned on 50 randomly selected classes from CIFAR-100 (Krizhevsky, 2009), and the task is to predict the training categories from model weights alone. To assess scalability to generative foundation models, we additionally use the Stable Diffusion (Rombach et al., 2022) LoRA (Hu et al., 2022; Ruiz et al., 2023) subset of Model Jungle in two settings—$SD_{200}$ (25 models per class over 200 ImageNet (Deng et al., 2009) classes) and $SD_{1k}$ (5 models per class over the full 1,000 classes)—evaluated in Sections 5.5 and 5.6.

**Baselines.** We compare MVProbe against state-of-the-art weight-space learning methods: (1) StatNN (Unterthiner et al., 2020), which represents models by concatenating seven statistics (mean, variance, and five quantiles) from model weights; (2) ProbeGen (Kahana et al., 2025), which introduces a shared probe generator to increase the expressivity of probe representations; (3) ProbeX (Horwitz et al., 2025), which extends to single-layer weight matrices, enabling scalability to larger models. Although graph-based and other equivariant weight-space approaches (Zhang et al., 2023; Navon et al., 2023; Kofinas et al., 2024; Lim et al., 2024) are possible baselines, they are excluded due to their prohibitive computational overhead on large weights and their tendency to diverge in single-layer probing (Horwitz et al., 2025). See Appendix L for direct comparison on small-scale benchmarks where they remain feasible.

**Implementation details.** We use $r = 128$ probes per branch, projection dimension $d = 128$, and output representation dimension 512. Models are trained with the Adam optimizer (Kingma & Ba, 2015) (learning rate $3 \times 10^{-4}$) for 500 epochs with batch size 128. All experiments use a single NVIDIA RTX 3090 GPU. We use the best-performing layers $\ell = 67, 59, 64, 47$ for ResNet, SupViT, MAE, and

---

[3] https://huggingface.co/ProbeX

DINO, respectively, while using $\ell = 46$ for all Stable Diffusion LoRA experiments. Full architecture details, hyperparameters, and the efficient Gram-matrix derivation are in Appendix A; the algorithm pseudocode is in Appendix G.

## 5.1. Main Results on Model Jungle

*Table 1.* **Model Jungle dataset classification accuracy.** We report accuracy (%) averaged over 5 seeds, evaluated at the best-performing layer for each model family. Best results are in **bold**.

| Method | ResNet | SupViT | MAE | DINO |
|---|---|---|---|---|
| StatNN | 55.20 | 55.80 | 54.83 | 55.69 |
| ProbeGen | 78.27 | 78.48 | 70.68 | 61.26 |
| ProbeX | 81.61 | 88.08 | 77.11 | 72.54 |
| ProbeX ($\times 4$) | 87.16 | 90.33 | 77.26 | 73.25 |
| MVProbe | **92.24** | **92.33** | **81.62** | **78.29** |
| *Improvement* | *+5.08* | *+2.00* | *+4.36* | *+5.04* |

As shown in Table 1, MVProbe consistently outperforms all baselines across all four architectures. Notably, we achieve substantial improvements over the previous state-of-the-art ProbeX ($\times 4$): +5.08% on ResNet, +2.00% on SupViT, +4.36% on MAE, and +5.04% on DINO at each model family's best-performing layer. The improvements are particularly pronounced on architectures where ProbeX struggled (ResNet, DINO), demonstrating that our multi-perspective probing effectively captures structural information that single-direction probe responses ($\mathbf{XU}$) miss.[4] Note that encoding model weights into hand-crafted statistics (StatNN) is insufficient to express large-scale models. Even a more advanced method like ProbeGen, proposed for probing across layers on smaller-scale settings, still underperforms on the large-scale Model Jungle benchmark, highlighting the challenge of performing classification from a single layer in large models.

## 5.2. Robustness to Layer Selection

Neural networks exhibit heterogeneous representations across layer depths. Consequently, performance can vary substantially across layers, making the search for an optimal layer computationally costly. In Figure 3, we compare the layer-wise performance of ProbeX and MVProbe, shown as dashed and solid curves, respectively. We focus on intermediate layers, indexing layers by $\ell$, and report results for $\ell \in \{20, \ldots, 73\}$, as very early or deep layers contain limited task-relevant information regardless of the fine-tuning objective. As indicated by the shaded bands, MVProbe consistently improves performance for

---

[4]MVProbe *uniformly dominates* ProbeX($\times 4$) under sufficient row-side rank; see Appendix B.3 for the formal definition and theory.

ResNet and SupViT while simultaneously reducing layer-wise variability. For MAE and DINO, although the performance fluctuations shrink only slightly, MVProbe still consistently improves performance across the majority of layers (See Section 6 for further discussion). We additionally mark the layer achieving the maximum per-layer gain $\text{Acc}_{\text{MVProbe}}(\ell) - \text{Acc}_{\text{ProbeX}}(\ell)$ for each model family. These markers serve as a concise summary of where the improvement peaks; importantly, MVProbe outperforms ProbeX over a broad span of intermediate layers rather than relying on a single optimal layer, reflecting improved robustness to layer selection.

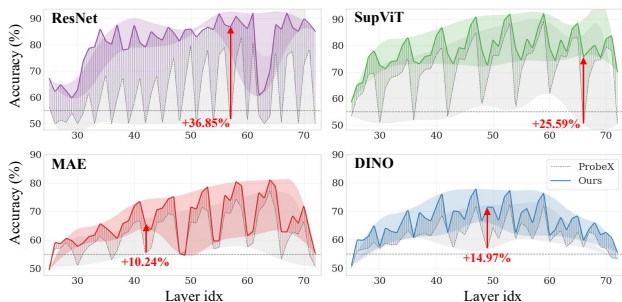

*Figure 3.* **Layer-wise performance comparison.** MVProbe (solid lines) vs. ProbeX (dashed lines) across all layers. Shaded bands indicate the performance volatility. MVProbe maintains higher accuracy across most layers and shows less sensitivity to layer selection.

## 5.3. Ablation Study: Branch Contributions

We analyze the contribution of each probing branch in Table 2. Each row shows the performance when using only the specified branch(es).

*Table 2.* **Ablation study on probing branches.** Each row shows accuracy (%) when using only the specified branch(es). $\mathbf{XU}$: row probing, $\mathbf{X}^\top \mathbf{V}$: column probing, $\mathbf{XX}^\top \mathbf{W}$: row kernel probing, $\mathbf{X}^\top \mathbf{XZ}$: column kernel probing. Best results are in **bold** and the second-best are underlined.

| Branch Configuration | ResNet | SupViT | MAE | DINO |
|---|---|---|---|---|
| $\mathbf{XU}$ only | 90.42 | 89.09 | 78.35 | 74.17 |
| $\mathbf{X}^\top \mathbf{V}$ only | 88.94 | 87.55 | 77.01 | 72.04 |
| $\mathbf{XX}^\top \mathbf{W}$ only | 89.26 | 89.52 | 78.28 | 72.68 |
| $\mathbf{X}^\top \mathbf{XZ}$ only | 87.00 | 90.56 | 78.22 | 72.35 |
| First-order ($\mathbf{XU}, \mathbf{X}^\top \mathbf{V}$) | 91.24 | 89.88 | 79.74 | 76.01 |
| Second-order ($\mathbf{XX}^\top \mathbf{W}, \mathbf{X}^\top \mathbf{XZ}$) | 90.22 | 92.04 | 80.57 | 75.82 |
| MVProbe (All four branches) | **92.24** | **92.33** | **81.62** | **78.29** |

The results reveal several insights:

**Each branch provides complementary information.** Combining all four branches yields the best performance across all architectures. No single branch alone matches the full model, confirming that first-order and second-order perspectives capture different structural aspects.

**Row vs. column asymmetry.** Row-derived branches tend to be stronger on ResNet and DINO at both orders (e.g., $\mathbf{XU} > \mathbf{X}^\top\mathbf{V}$ and $\mathbf{XX}^\top\mathbf{W} > \mathbf{X}^\top\mathbf{XZ}$). MAE shows the same tendency at first order, while its second-order row/column kernels are nearly tied (78.28% vs. 78.22%). SupViT is order-dependent: $\mathbf{XU}$ gives a higher score (89.09% vs. 87.55%), but $\mathbf{X}^\top\mathbf{XZ}$ gives a higher score (90.56% vs. 89.52%). Overall, the more informative axis (row vs. column) depends on both the backbone and the interaction order.

**Complementarity across orders.** The second-order combination is not consistently better than the first-order one (it is slightly lower on DINO), yet aggregating all four branches achieves the best accuracy across all architectures (Table 2). This suggests that second-order correlation structure provides information that complements direct first-order responses, and is most effective when combined rather than used in isolation.

**Probe count vs. branch design.** Consistent with this, increasing the number of ProbeX probes alone does not improve performance (Table 1), highlighting that the gains primarily come from the proposed multi-branch design.

### 5.4. Ablation Study: Standardization

Theorem 4.3 suggests that multi-order branches can exhibit scale imbalance, so that naive concatenation may overweight second-order responses. We perform an ablation by comparing MVProbe with and without per-sample standardization of branch features prior to aggregation.

*Table 3.* **Effect of Standardization.** Accuracy (%) of MVProbe with and without per-sample standardization across model families. $\Delta$ denotes the mean per-layer gain (w/ Std $-$ w/o Std), computed from unrounded values, and % Improved denotes the fraction of layers with $\Delta > 0$. Standardization consistently improves performance, with 89.2% of layers showing improvement (one-sided sign test, $p < 0.001$).

| Model | w/o Std | w/ Std (MVProbe) | $\Delta$ | % Improved |
|---|---|---|---|---|
| DINO | 58.9 | 63.0 | **+4.2** | 97.3 |
| MAE | 62.1 | 62.9 | +0.8 | 76.7 |
| ResNet | 69.7 | 73.8 | **+4.1** | 96.2 |
| SupViT | 71.4 | 73.0 | +1.6 | 83.8 |
| **Average** | 65.9 | 68.8 | **+2.8** | **89.2** |

In Table 3, standardization yields a mean gain of +2.8% across all models (one-sample $t$-test on per-layer $\Delta$, $p < 0.001$), and improves 89.2% of layers (one-sided sign test, $p < 0.001$). Gains are largest for DINO (+4.2%) and ResNet (+4.1%), while MAE shows smaller improvements (+0.8%). Gains also vary with depth (Table 4; Kruskal–Wallis on $\Delta$: ViT-based $p < 0.01$, ResNet $p < 0.001$), with the largest average improvements in middle layers. Overall, these results are consistent with Corollary 4.4, suggesting that standardization helps balance multi-branch contributions. Full

*Table 4.* **Standardization effect by layer depth.** Gains vary with depth; middle layers show the largest average improvements in both ViT-based models (blocks 4–7) and ResNet (stage 2). $\Delta$ is computed from unrounded per-layer means.

| Model | Layer Depth | w/o Std | w/ Std | $\Delta$ |
|---|---|---|---|---|
| ViT (All) | Early (block 0–3) | 60.1 | 62.2 | +2.2 |
| | Middle (block 4–7) | 70.6 | 73.4 | **+2.9** |
| | Late (block 8–11) | 62.8 | 64.4 | +1.6 |
| | *Average* | *64.5* | *66.7* | *+2.2* |
| ResNet | Early (stage 0–1) | 54.0 | 55.9 | +1.8 |
| | Middle (stage 2) | 73.6 | 78.6 | **+5.1** |
| | Late (stage 3) | 80.7 | 83.6 | +2.9 |
| | *Average* | *69.9* | *74.0* | *+4.1* |

*Note:* ViT (All) includes SupViT, MAE, and DINO.

test details and grouping conventions are provided in Appendix E.3, and a comparison against alternative fusion strategies (per-branch $L_2$ normalization versus per-sample standardization, and simple concatenation versus learned branch weighting) is reported in Table 24 in Appendix J. Per-sample standardization combined with simple concatenation remains the most robust choice across architectures.

### 5.5. Scaling to Generative Foundation Models

Unlike fully fine-tuned models, LoRA checkpoints expose only a small fraction of the foundation model's weight updates, making property prediction substantially more challenging. The two methods are comparable on $SD_{200}$, but on the harder $SD_{1k}$, ProbeX collapses to 35.75% in-distribution accuracy while MVProbe retains 97.88% (Table 5); the gap holds across layers (Figure 4). On the failure layers where ProbeX drops below 26%, MVProbe maintains 62–90% accuracy (Appendix K).

*Table 5.* **SD LoRA classification accuracy.** We report mean accuracy (%) with standard deviation (subscript) over 5 seeds on Stable Diffusion LoRA checkpoints at layer 46, evaluated under In-Distribution (training classes) and Zero-shot (held-out classes) protocols. Best results are in **bold**.

| | Method | In-Dist. | Zero-shot |
|---|---|---|---|
| $SD_{200}$ | ProbeX | $98.48_{\pm 0.48}$ | $94.01_{\pm 0.77}$ |
| | MVProbe | $\mathbf{99.80_{\pm 0.00}}$ | $\mathbf{95.53_{\pm 0.65}}$ |
| $SD_{1k}$ | ProbeX | $35.75_{\pm 2.44}$ | $52.42_{\pm 2.48}$ |
| | MVProbe | $\mathbf{97.88_{\pm 0.37}}$ | $\mathbf{97.96_{\pm 0.29}}$ |

### 5.6. Generalization to Broader Weight-Space Tasks

To further validate the generality of MVProbe beyond closed-set classification, we evaluate two additional tasks on SD LoRA: $k$-nearest-neighbor (kNN) retrieval and open-set classification via one-class classification (OCC). These

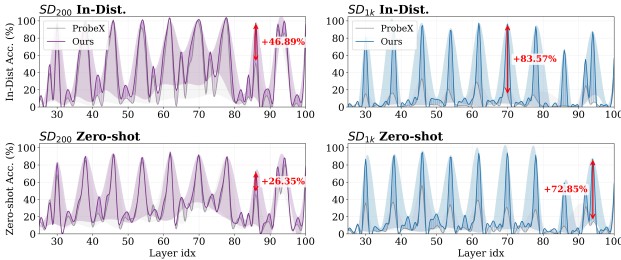

*Figure 4.* **Layer-wise performance on SD LoRA.** MVProbe (solid lines) vs. ProbeX (dashed lines) across all layers on $SD_{200}$ and $SD_{1k}$ (In-Distribution and Zero-shot). Shaded bands indicate per-layer volatility; red arrows mark the largest gains.

*Table 6.* **kNN retrieval and OCC accuracy on SD LoRA.** We report accuracy (%) across different $k$ values on $SD_{200}$ and $SD_{1k}$ at layer 46. OCC at $k$=5 on $SD_{1k}$ is undefined since each class contains only 5 models. Best results are in **bold**.

| | kNN | | | | OCC | | | |
| | $SD_{200}$ | | $SD_{1k}$ | | $SD_{200}$ | | $SD_{1k}$ | |
| $k$ | ProbeX | MVProbe | ProbeX | MVProbe | ProbeX | MVProbe | ProbeX | MVProbe |
|---|---|---|---|---|---|---|---|---|
| 1 | 98.59 | **99.80** | 21.24 | **93.99** | 81.46 | **100.0** | 73.20 | **99.99** |
| 2 | 93.99 | **99.80** | 17.64 | **91.18** | 81.55 | **100.0** | 77.49 | **99.73** |
| 5 | 92.38 | **99.80** | 13.23 | **87.98** | 81.34 | **100.0** | – | – |

tasks test whether the learned weight-space representations support retrieval and generalize to unseen classes. As shown in Table 6, MVProbe consistently and substantially outperforms ProbeX. In kNN retrieval, ProbeX degrades sharply on $SD_{1k}$, dropping to 21.24% at $k$=1 and 13.23% at $k$=5, while MVProbe retains 93.99% and 87.98% respectively. A similar gap is observed for OCC, where MVProbe reaches near-perfect 100% on $SD_{200}$ across all $k$, while ProbeX plateaus around 81%. These results suggest that the richer structural information captured by multi-view probing yields representations that are not only more discriminative but also more geometrically coherent in weight space.

## 6. Discussion

*Table 7.* **Comparison of FLOPs and training time per epoch.**

| Model | FLOPs | Training time (sec) |
|---|---|---|
| ProbeX | 97M | 1.278 |
| ProbeX ($\times$4) | 383M | 1.376 |
| MVProbe | 537M | 1.375 |

**Computational overhead.** One might question the significant computational overhead of MVProbe due to its second-order operations. Let $m$ and $n$ be the dimensions of the weight matrix and $r$ be the number of probes. The first-order branches compute $\mathbf{X}\mathbf{U}$ and $\mathbf{X}^\top\mathbf{V}$, each requiring $O(mnr)$ operations. For the second-order branches, a naive implementation of $\mathbf{X}\mathbf{X}^\top\mathbf{W}$ would incur $O(m^2n)$ cost from explicitly forming the Gram matrix; however, we compute it associatively as $\mathbf{X}(\mathbf{X}^\top\mathbf{W})$ in $O(mnr)$, and similarly

compute $\mathbf{X}^\top\mathbf{X}\mathbf{Z}$ as $\mathbf{X}^\top(\mathbf{X}\mathbf{Z})$ in $O(mnr)$. Thus MVProbe retains $O(mnr)$ complexity overall, linear in the matrix dimensions and the probe budget. Consistent with this analysis, Table 7 shows that although MVProbe involves substantially more FLOPs than ProbeX, the actual training time per epoch remains comparable across methods, indicating that these operations are efficiently parallelized on modern GPUs. Full runtime, throughput, and peak-memory profiling across architectures, including SD LoRA at $1280 \times 1280$, is reported in Table 23 in Appendix I.

**Layer selection for ViT-based models.** Empirically, intermediate-depth layers are generally most reliable across architectures, and for ViT-based models the value projection weights in attention blocks tend to provide more stable accuracy than other weight types; see Appendix E for detailed layer-wise results.

**Higher-order branches.** Extending MVProbe to third- and fourth-order responses yields architecture-dependent results: gains on ResNet (+1.4%) and SupViT (+1.0%), degradation on DINO ($-1.9\%$), and only marginal change on MAE ($-0.5\%$). We therefore adopt the 4-branch (first+second order) design as default and treat higher orders as optional; full ablation in Appendix D.1.

**Limitations and future work.** Despite consistent gains, absolute accuracy remains relatively low on MAE and DINO under the single-layer probing setting. Future work includes understanding which layer types and depths are most error-prone, how weight statistics (e.g., scale, anisotropy, or rank structure) correlate with performance drops, and exploring remedies such as multi-layer aggregation or architecture-aware branch selection.

## 7. Conclusion

In this work, we studied the limitations of existing probing approaches for weight-space learning. We theoretically characterized a probing ambiguity problem where distinct weight matrices yield identical first-order representations. Motivated by this analysis, we proposed MVProbe, a multi-view probing framework that jointly captures first-order structure and second-order (Gram-based) interactions. By incorporating kernel-based probing and per-sample standardization, MVProbe addresses the ambiguity of prior methods while remaining computationally efficient and scalable to large models. We also provided a unified interpretation through the lenses of kernel methods and landmark-based representations, clarifying why interaction-aware probing is helpful for reliable weight-based model identification. Extensive experiments on the Model Jungle benchmark show that MVProbe consistently outperforms baselines across diverse architectures and layers, including regimes where prior methods degrade toward chance-level performance.

## Impact Statement

This work contributes to weight-space learning, supporting legitimate use cases such as model identification, attribution, and quality control in large open-source repositories. The ability to distinguish models from their weights also carries potential risks (e.g., unsanctioned identification of fine-tuned derivatives), but MVProbe operates only on voluntarily shared public checkpoints and does not recover training data. We do not foresee novel ethical concerns beyond those already present in the broader weight-space learning literature.

## Acknowledgments

This work was supported by the Institute of Information & Communications Technology Planning & Evaluation (IITP) grants funded by the Korea government (MSIT): (No.RS-2020-II201336, Artificial Intelligence Graduate School Program (UNIST)), (No.RS-2022-II220264, Comprehensive Video Understanding and Generation with Knowledge-based Deep Logic Neural Network), (RS-2025-25442149), (RS-2025-25442824, AI Star Fellowship Program (Ulsan National Institute of Science and Technology)), (RS-2026-25528781, Hyper-scale Industrial AI Research Support (R&D) Program, Development of an industry-specified intelligent data processing and federated learning platform), the National Research Foundation of Korea (NRF) grant funded by the Korea government (MSIT) (No.2022R1C1C100849612), and the InnoCORE program of the Ministry of Science and ICT (N10250156).

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

# A. Implementation Details

## A.1. Architecture Details

**Probe Matrices.** Each of the four branches uses its own learnable probe matrix:

- Row probing: $\mathbf{U} \in \mathbb{R}^{n \times r}$

- Column probing: $\mathbf{V} \in \mathbb{R}^{m \times r}$

- Row kernel probing: $\mathbf{W} \in \mathbb{R}^{m \times r}$

- Column kernel probing: $\mathbf{Z} \in \mathbb{R}^{n \times r}$

All probe matrices are initialized using Xavier uniform initialization.

**Projection Layers.** Each branch has a dedicated projection layer that maps the flattened probe response to a $d$-dimensional vector:

- Row probing: $\pi_1 : \mathbb{R}^{m \times r} \to \mathbb{R}^d$

- Column probing: $\pi_2 : \mathbb{R}^{n \times r} \to \mathbb{R}^d$

- Row kernel probing: $\pi_3 : \mathbb{R}^{m \times r} \to \mathbb{R}^d$

- Column kernel probing: $\pi_4 : \mathbb{R}^{n \times r} \to \mathbb{R}^d$

Each projection layer is a single linear layer followed by ReLU activation.

**Classification Head.** The concatenated features from all four branches (dimension $4d$) are passed through a single linear layer to produce class logits.

## A.2. Probe Response Standardization

Before projection, we apply per-sample standardization to each probe response matrix:

$$\tilde{\mathbf{S}} = \frac{\mathbf{S} - \mu_{\mathbf{S}}}{\sigma_{\mathbf{S}} + \epsilon}, \tag{19}$$

where $\mu_{\mathbf{S}}$ and $\sigma_{\mathbf{S}}$ are the mean and standard deviation computed over all entries of $\mathbf{S}$ for each sample in the batch, and $\epsilon = 10^{-8}$ prevents division by zero. This standardization ensures consistent input scales regardless of the weight matrix magnitude or architecture.

## A.3. Efficient Gram Matrix Computation

A naive implementation of the second-order branches would first compute the full Gram matrix ($O(m^2n)$ or $O(n^2m)$), then multiply by the probe matrix. We avoid this by exploiting associativity:

$$\mathbf{X}\mathbf{X}^\top \mathbf{W} = \mathbf{X}(\mathbf{X}^\top \mathbf{W}), \tag{20}$$
$$\mathbf{X}^\top \mathbf{X}\mathbf{Z} = \mathbf{X}^\top (\mathbf{X}\mathbf{Z}). \tag{21}$$

Each computation now requires $O(mnr)$ operations, the same complexity as first-order branches. This makes the second-order branches computationally efficient even for large weight matrices.

## A.4. Hyperparameters

Table 8 summarizes the hyperparameters used in our experiments.

*Table 8.* Hyperparameters used in MVProbe.

| Hyperparameter | Value |
|---|---|
| Probes per branch ($r$) | 128 |
| Projection dim ($d$) | 128 |
| Batch size | 128 |
| Learning rate | $3 \times 10^{-4}$ |
| Weight decay | $10^{-5}$ |
| Epochs | 500 |
| Optimizer | Adam |
| Loss | BCE |

# B. Proofs of Theoretical Results

## B.1. Proof of Theorem 4.1 (Expressiveness of Second-Order Probes)

*Proof.* Since $\text{rank}(\mathbf{U}) = r < n$, the left nullspace of $\mathbf{U}$ is nontrivial, so there exists $\mathbf{w} \neq \mathbf{0} \in \mathbb{R}^n$ such that $\mathbf{w}^\top \mathbf{U} = \mathbf{0}^\top$.

Choose any $\mathbf{v} \in \mathbb{R}^n$ such that $\mathbf{v}^\top \mathbf{U} \neq \mathbf{0}^\top$ (such a $\mathbf{v}$ exists unless $\mathbf{U} = \mathbf{0}$, which is excluded by $\text{rank}(\mathbf{U}) = r$). Pick any $\mathbf{a} \neq \mathbf{0} \in \mathbb{R}^m$ and define:

$$\mathbf{X}_1 := \mathbf{a}\mathbf{v}^\top, \qquad \mathbf{X}_2 := \mathbf{a}(\mathbf{v} + \mathbf{w})^\top. \tag{22}$$

**Step 1: First-order features are identical.**

$$\begin{aligned} \mathbf{X}_2\mathbf{U} &= \mathbf{a}(\mathbf{v} + \mathbf{w})^\top \mathbf{U} \\ &= \mathbf{a}\mathbf{v}^\top \mathbf{U} + \mathbf{a}\mathbf{w}^\top \mathbf{U} = \mathbf{X}_1\mathbf{U}, \end{aligned} \tag{23}$$

since $\mathbf{w}^\top \mathbf{U} = \mathbf{0}^\top$.

**Step 2: Second-order features differ.** For any rank-one matrix $\mathbf{X} = \mathbf{a}\mathbf{p}^\top$, we have $\mathbf{X}^\top \mathbf{X} = \|\mathbf{a}\|_2^2 \, \mathbf{p}\mathbf{p}^\top$. Therefore:

$$(\mathbf{X}_1^\top \mathbf{X}_1)\mathbf{U} = \|\mathbf{a}\|_2^2 \, \mathbf{v}\mathbf{v}^\top \mathbf{U}, \tag{24}$$
$$(\mathbf{X}_2^\top \mathbf{X}_2)\mathbf{U} = \|\mathbf{a}\|_2^2 \, (\mathbf{v}+\mathbf{w})(\mathbf{v}+\mathbf{w})^\top \mathbf{U}. \tag{25}$$

Taking the difference and expanding:

$$(\mathbf{X}_2^\top \mathbf{X}_2)\mathbf{U} - (\mathbf{X}_1^\top \mathbf{X}_1)\mathbf{U} \tag{26}$$
$$= \|\mathbf{a}\|_2^2 (\mathbf{w}\mathbf{v}^\top \mathbf{U} + \mathbf{v}\mathbf{w}^\top \mathbf{U} + \mathbf{w}\mathbf{w}^\top \mathbf{U}) \tag{27}$$
$$= \|\mathbf{a}\|_2^2 \, \mathbf{w}\mathbf{v}^\top \mathbf{U} \neq \mathbf{0}, \tag{28}$$

where we used $\mathbf{w}^\top \mathbf{U} = \mathbf{0}^\top$ to eliminate the last two terms. Since $\mathbf{w} \neq \mathbf{0}$ and $\mathbf{v}^\top \mathbf{U} \neq \mathbf{0}^\top$, their outer product is nonzero. $\square$

### B.2. Proof of Theorem 4.2 (Transpose-Complement Non-Redundancy)

*Proof.* Since $\mathrm{rank}(\mathbf{U}) = r < n$, there exists $\mathbf{w} \in \mathbb{R}^n \setminus \{\mathbf{0}\}$ with $\mathbf{w}^\top \mathbf{U} = \mathbf{0}^\top$. Choose $\mathbf{a} \in \mathbb{R}^m$ such that $\mathbf{V}^\top \mathbf{a} \neq \mathbf{0}$ (possible since $\mathbf{V} \neq \mathbf{0}$). Define

$$\mathbf{X}_2 = \mathbf{X}_1 + \mathbf{a}\mathbf{w}^\top.$$

Then $\mathbf{X}_2 \mathbf{U} = \mathbf{X}_1 \mathbf{U} + \mathbf{a}(\mathbf{w}^\top \mathbf{U}) = \mathbf{X}_1 \mathbf{U}$, while

$$\mathbf{X}_2^\top \mathbf{V} - \mathbf{X}_1^\top \mathbf{V} = \mathbf{w}\,\mathbf{a}^\top \mathbf{V} \neq \mathbf{0},$$

so the column-side response distinguishes the two matrices that are indistinguishable on the row side. $\square$

### B.3. Uniform Domination of ProbeX($\times 4$)

Below we formalize the relation between MVProbe and ProbeX($\times 4$) under fixed probes.

**Definition (Uniform Domination).** Let $H(\mathbf{X}) = \mathbf{X}\widetilde{\mathbf{U}}$ be the ProbeX($\times 4$) feature with stacked first-order bank $\widetilde{\mathbf{U}} \in \mathbb{R}^{n \times 4r}$, and let $G(\mathbf{X}) = (\mathbf{X}\mathbf{U}, \mathbf{X}^\top \mathbf{V}, \mathbf{X}\mathbf{X}^\top \mathbf{W}, \mathbf{X}^\top \mathbf{X}\mathbf{Z})$ be the MVProbe feature. We say MVProbe *uniformly dominates* ProbeX($\times 4$) if $G(\mathbf{X}_1) = G(\mathbf{X}_2) \Rightarrow H(\mathbf{X}_1) = H(\mathbf{X}_2)$ for all $\mathbf{X}_1, \mathbf{X}_2$.

**Proposition (Sufficient Condition).** If $\mathrm{Col}(\widetilde{\mathbf{U}}) \subseteq \mathrm{Col}(\mathbf{U})$, then MVProbe uniformly dominates ProbeX($\times 4$).

*Proof.* Assume $\mathrm{Col}(\widetilde{\mathbf{U}}) \subseteq \mathrm{Col}(\mathbf{U})$. Then there exists a matrix $\mathbf{C}$ such that $\widetilde{\mathbf{U}} = \mathbf{U}\mathbf{C}$. Now suppose $G(\mathbf{X}_1) = G(\mathbf{X}_2)$. Since $\mathbf{X}\mathbf{U}$ is the first component of $G$, we have $\mathbf{X}_1 \mathbf{U} = \mathbf{X}_2 \mathbf{U}$. Therefore,

$$H(\mathbf{X}_1) = \mathbf{X}_1 \widetilde{\mathbf{U}} = \mathbf{X}_1 \mathbf{U}\mathbf{C} = (\mathbf{X}_1 \mathbf{U})\mathbf{C}$$
$$= (\mathbf{X}_2 \mathbf{U})\mathbf{C} = \mathbf{X}_2 \mathbf{U}\mathbf{C} = \mathbf{X}_2 \widetilde{\mathbf{U}} = H(\mathbf{X}_2).$$

Thus $G(\mathbf{X}_1) = G(\mathbf{X}_2) \Rightarrow H(\mathbf{X}_1) = H(\mathbf{X}_2)$, which is exactly uniform domination of ProbeX($\times 4$) by MVProbe. $\square$

**Corollary (Full-Rank XU Branch Subsumes Any First-Order Bank).** If $\mathrm{rank}(\mathbf{U}) = n$, then $\mathrm{Col}(\mathbf{U}) = \mathbb{R}^n$, so any ProbeX($\times 4$) bank $\widetilde{\mathbf{U}} \in \mathbb{R}^{n \times 4r}$ satisfies $\mathrm{Col}(\widetilde{\mathbf{U}}) \subseteq \mathrm{Col}(\mathbf{U})$ trivially. Hence MVProbe uniformly dominates every ProbeX($\times 4$) implementation in this regime; once the row-side first-order branch is sufficiently rich, additional first-order probes can no longer add recoverable information, and further gains must come from the complementary branches $\mathbf{X}^\top \mathbf{V}$ and the Gram-based second-order branches.

### B.4. Proof of Theorem 4.3 (Scale Imbalance)

*Proof.* **First-order branch scale.** Write $\mathbf{S}^{(1)} = \mathbf{X}\mathbf{U}$ where $\mathbf{U}$ has unit-norm columns. For entry $(i, k)$:

$$(\mathbf{X}\mathbf{U})_{ik} = \sum_{j=1}^n X_{ij} U_{jk}.$$

Since entries $X_{ij}$ are i.i.d. with mean 0 and variance $\sigma^2$:

$$\mathbb{E}\big[(\mathbf{X}\mathbf{U})_{ik}^2\big] = \sum_{j=1}^n U_{jk}^2 \, \mathbb{E}[X_{ij}^2] = \sigma^2 \|\mathbf{U}_{:,k}\|_2^2 = \sigma^2.$$

Summing over all entries:

$$\mathbb{E}\|\mathbf{S}^{(1)}\|_F^2 = m \cdot r \cdot \sigma^2.$$

**Second-order branch scale.** Let $\mathbf{G} = \mathbf{X}\mathbf{X}^\top \in \mathbb{R}^{m \times m}$. We need $\mathbb{E}[\mathbf{G}^2]$.

Write $\mathbf{X} = [\mathbf{x}_1, \dots, \mathbf{x}_n]$ as columns, where $\mathbf{x}_\ell \sim \mathcal{N}(0, \sigma^2 \mathbf{I}_m)$ are independent. Then $\mathbf{G} = \sum_{\ell=1}^n \mathbf{x}_\ell \mathbf{x}_\ell^\top$.

For the diagonal term $(\mathbf{x}\mathbf{x}^\top)^2 = \|\mathbf{x}\|_2^2 \mathbf{x}\mathbf{x}^\top$. By rotational symmetry:

$$\mathbb{E}\big[(\mathbf{x}\mathbf{x}^\top)^2\big] = (m+2)\sigma^4 \mathbf{I}.$$

For cross terms ($\ell \neq k$):

$$\mathbb{E}\big[(\mathbf{x}_\ell \mathbf{x}_\ell^\top)(\mathbf{x}_k \mathbf{x}_k^\top)\big] = \mathbb{E}[\mathbf{x}_\ell \mathbf{x}_\ell^\top]\mathbb{E}[\mathbf{x}_k \mathbf{x}_k^\top] = \sigma^4 \mathbf{I}.$$

Combining:

$$\mathbb{E}[\mathbf{G}^2] = n(m+2)\sigma^4 \mathbf{I} + n(n-1)\sigma^4 \mathbf{I}$$
$$= n(n+m+1)\sigma^4 \mathbf{I}. \tag{29}$$

Therefore:

$$\mathbb{E}\|\mathbf{S}^{(2)}\|_F^2 = \mathrm{tr}(\mathbf{W}^\top \mathbb{E}[\mathbf{G}^2]\mathbf{W}) = n(n+m+1)\sigma^4 \cdot r.$$

The ratio follows by division. $\square$

## C. Connection to Kernel Methods

Our second-order probing branches have a natural interpretation through the lens of kernel methods. Given a weight matrix $\mathbf{X} \in \mathbb{R}^{m \times n}$, the row kernel matrix $\mathbf{K}_{\mathrm{row}} = \mathbf{X}\mathbf{X}^\top$ defines a linear kernel on the rows of $\mathbf{X}$.

**Kernel PCA Connection.** In kernel PCA (Schölkopf et al., 1998), one computes the eigenvectors of the centered kernel matrix to find principal components in the feature space. Our row kernel probing $\mathbf{X}\mathbf{X}^\top \mathbf{W}$ can be viewed as learning *task-specific* directions in this kernel space, rather than using the data-agnostic principal components.

**Random Features Connection.** Random features (Rahimi & Recht, 2007) approximate kernel functions by projecting data onto random directions. Our learned probe vectors $\mathbf{W}$ serve a similar role, but are optimized for the downstream classification task rather than approximating a specific kernel.

**Why Kernel Probing Helps.** The kernel matrix $\mathbf{XX}^\top$ encodes pairwise relationships that the original matrix $\mathbf{X}$ does not directly expose. Entry $(i, j)$ of the kernel matrix measures the similarity between rows $i$ and $j$. By probing through this matrix, we extract features that capture how rows relate to each other—information that first-order probing $\mathbf{XU}$ cannot access since it treats each row independently.

# D. Additional Experimental Results

## D.1. Higher-Order Branch Ablation

We extend MVProbe (up to 2nd order; 4 branches) by adding third-order (6 branches) and fourth-order (8 branches) responses. The third-order branches use $\mathbf{XX}^\top\mathbf{X}$ and $\mathbf{X}^\top\mathbf{XX}^\top$; the fourth-order branches use $(\mathbf{XX}^\top)^2$ and $(\mathbf{X}^\top\mathbf{X})^2$. As shown in Table 9, the impact is architecture-dependent: third–fourth order branches improve ResNet and SupViT (peaking at the 8-branch setting) but decrease DINO and yield only marginal changes on MAE. We therefore adopt the 4-branch (first+second order) design as the default for robust performance.

*Table 9.* **Effect of higher-order extensions.** Best results are in **bold** and the second-best are underlined.

| Order (branches) | ResNet | SupViT | MAE | DINO |
|---|---|---|---|---|
| MVProbe | 92.24 | 92.33 | **81.62** | **78.29** |
| + up to 3rd (6) | 93.15 | 92.36 | 80.63 | 77.02 |
| + up to 4th (8) | **93.69** | **93.34** | 81.16 | 76.38 |

## D.2. Detailed Per-Architecture Analysis

Table 10 provides a detailed breakdown of results across all architectures and layer configurations.

*Table 10.* Detailed accuracy comparison across architectures.

| Arch | Layer | ProbeX | MVProbe | Δ |
|---|---|---|---|---|
| ResNet | 59 | 83.23 | 88.40 | +5.17 |
| SupViT | 59 | 89.42 | 92.11 | +2.69 |
| MAE | 64 | 77.14 | 81.62 | +4.48 |
| DINO | 59 | 70.50 | 76.48 | +5.98 |
| **Avg** | | 80.07 | 84.65 | +4.58 |

## D.3. Win Rate Across All Layers

We evaluate MVProbe against ProbeX across all available layers in each architecture. Table 11 shows the win rate (percentage of layers where MVProbe outperforms ProbeX).

*Table 11.* Win rate across all layers.

| Architecture | # Layers | Win Rate (%) |
|---|---|---|
| ResNet101 | 105 | 94.3 |
| SupViT | 74 | 95.9 |
| MAE | 74 | 93.2 |
| DINO | 74 | 97.3 |
| **Total** | 327 | 95.1 |

MVProbe outperforms ProbeX on 95.1% of all layers (311 out of 327), demonstrating consistent improvements across the entire layer spectrum.

# E. Architectural Analysis

## E.1. Performance Analysis Over Depth

*Table 12.* Stage-wise probing performance of ResNet architecture stages.

| **ResNet (Layer Indices)** | **Mean Acc. (%)** |
|---|---|
| Stage 0 (0–12) | 54.18 |
| Stage 1 (13–24) | 58.59 |
| Stage 2 (25–61) | **80.76** |
| Stage 3 (62–104) | 77.98 |

*Table 13.* Block-wise probing performance of ViT-based models (SupViT, MAE, and DINO). Each model is divided into three equal regimes: Early (0–24), Mid (25–49), and Late (50–74).

| | **Mean Accu. (%)** | | |
|---|---|---|---|
| | **DINO** | **MAE** | **SupViT** |
| Early (0–24) | 56.71 | 55.67 | 62.22 |
| Mid (25–49) | 66.14 | 63.56 | 76.27 |
| Late (50–74) | 66.39 | **69.49** | **80.87** |

To further investigate the information capacity of different architectural depths, we analyze the average accuracy across functionally identical layers for ResNet and ViT-based models. As demonstrated in Table 12 and Table 13, the trajectory of performance improvement varies significantly between convolutional and transformer-based designs. As shown in Table 12, the performance of the ResNet model exhibits a significant jump as the depth increases. While the early stages (Stage 0 and 1) maintain a modest accuracy, Stage 2 and 3 achieve the peak performance of 80.76% and 77.98%.

This suggests that the features extracted in the middle-to-late layers of the convolution architecture are the most separable and semantically rich for the weight classification. Table 13 illustrates the performance of SupViT, MAE, and DINO across three blocks: Early, Mid, and Late. SupViT remarkably outperforms other self-supervised counterparts (DINO and MAE) across all blocks. Both MAE and SupViT achieve their highest accuracy in the Late regime (Blocks 50–74), confirming that Transformer architectures successfully refine global context and semantic depth as the layers progress.

## E.2. Performance Analysis Over Functional Layers

*Table 14.* Analysis of accuracy deltas ($\Delta$) across all models on MVProbe, grouped by functionally identical layers. $\mu(\Delta)$ represents the average performance gain/loss per layer type.

| Architecture | Layer Type | $\mu(\Delta)$ (%) |
|---|---|---|
| ResNet | 1×1 Conv (down) | -4.21 |
| | 3×3 Conv | +0.34 |
| | 1×1 Conv (up) | **+4.36** |
| | Shortcut (Skip) | -0.26 |
| SupViT | Attention key / query | +1.36 |
| | Attention value ($W_V$) | **+6.23** |
| | Attention Output | +1.47 |
| | Intermediate MLP | -2.61 |
| MAE | Attention key / query | +2.8 |
| | Attention value ($W_V$) | **+3.76** |
| | Attention Output | -1.74 |
| | Intermediate MLP | -3.22 |
| DINO | Attention key / query | -0.51 |
| | Attention value ($W_V$) | **+2.72** |
| | Attention Output | +1.56 |
| | Intermediate MLP | -0.36 |

To understand the specific contributions of various architectural components, we categorize layers by their functional roles and calculate the mean accuracy delta ($\mu(\Delta)$) on MVProbe. The delta represents the performance gain or loss relative to the preceding layer. The comprehensive results are summarized in Table 14. We sort the performances across the functional layers as 1) 1x1 convolution for downsampling the channel, 2) 3x3 convolution, 3) 1x1 convolution for upsampling the channel for ResNet. For the case of ViT-based architectures, we decompose as 1) key / query layer in attention block, 2) value layer, 3) MLP at the end of attention block, 4) MLP layer at inter-blocks. ResNet architecture demonstrates a distinct functional hierarchy. The highest gain (+4.36%) is observed in the 1×1 Conv (up) layers, which are responsible for projecting features into a higher-dimensional space before the residual summation. This suggests that these expansion layers are

critical for creating separable representations. All ViT variants (SupViT, MAE, DINO) demonstrate consistent improvement on Attention Value ($W_V$) layers, with +6.23%, +3.76%, +2.72%, respectively. This suggests that while the Key and Query matrices manage spatial routing, the Value matrix serves as the primary repository for task-relevant semantic features that could be used for the weight classification task.

## E.3. Statistical Details for Standardization Ablation

*Table 15.* **Overall statistical tests for standardization.** $\Delta_\ell$ denotes per-layer accuracy gain (pp).

| Statistic | Value |
|---|---|
| # layers ($N$) | 325 |
| Mean gain ($\overline{\Delta}$) | 2.816 pp |
| Std. ($s_\Delta$) | 3.230 pp |
| 95% CI for $\overline{\Delta}$ | [2.463, 3.168] pp |
| One-sample $t$-test on $\Delta$ | $t(324) = 15.715$, $p = 4.62\times10^{-42}$ |
| Positive-gain layers | 290/325 (89.23%) |
| Sign test (one-sided) | $p = 1.98\times10^{-51}$ |

*Table 16.* **Per-model statistical tests.** One-sided sign tests are applied to $\Delta_\ell$.

| Model | $N$ | $\overline{\Delta}$ (pp) | $k/N$ (%) | $p$ |
|---|---|---|---|---|
| DINO | 73 | 4.158 | 71/73 (97.26) | $2.86\times10^{-19}$ |
| MAE | 73 | 0.822 | 56/73 (76.71) | $2.63\times10^{-6}$ |
| ResNet | 105 | 4.140 | 101/105 (96.19) | $1.23\times10^{-25}$ |
| SupViT | 74 | 1.579 | 62/74 (83.78) | $1.43\times10^{-9}$ |

*Table 17.* **Depth-group statistics for standardization gains.** $\overline{\Delta}$ is the mean gain (pp) and $k/N$ counts layers with $\Delta_\ell > 0$.

| Family | Depth group | $N$ | $\overline{\Delta}$ (pp) | $k/N$ (%) |
|---|---|---|---|---|
| ViT-based | Early (0–3) | 72 | 2.166 | 69/72 (95.83) |
| | Middle (4–7) | 72 | 2.856 | 62/72 (86.11) |
| | Late (8–11) | 70 | 1.620 | 53/70 (75.71) |
| | KW on $\Delta$ | $H(2) = 10.849$, $p = 4.41\times10^{-3}$ | | |
| ResNet | Early (0–1) | 23 | 1.843 | 22/23 (95.65) |
| | Middle (2) | 70 | 5.070 | 69/70 (98.57) |
| | Late (3) | 10 | 2.924 | 8/10 (80.00) |
| | KW on $\Delta$ | $H(2) = 16.934$, $p = 2.10\times10^{-4}$ | | |

**Setup and notation.** For each layer $\ell$, let $a_\ell^{\text{w/o}}$ and $a_\ell^{\text{w/}}$ denote accuracies (%) without and with per-sample standardization. We define the per-layer gain (percentage points) as

$$\Delta_\ell = a_\ell^{\text{w/}} - a_\ell^{\text{w/o}}. \tag{30}$$

All tests are one-sided, aligned with the hypothesis that standardization improves performance.

**Overall effect.** Table 15 summarizes the aggregate statistics. Across all models ($N = 325$), the mean gain is $\overline{\Delta} = 2.816$ pp with sample standard deviation $s_\Delta = 3.230$ pp. A one-sample $t$-test on $\Delta_\ell$ gives $t(324) = 15.715$, $p = 4.62 \times 10^{-42}$, with 95% CI [2.463, 3.168] pp. Moreover, $k = 290$ layers have $\Delta_\ell > 0$ (89.23%), and a one-sided sign test gives $p = 1.98 \times 10^{-51}$.

**By model family.** Table 16 reports layer counts, mean gains, and one-sided sign-test $p$-values per family.

**Depth-dependent effects.** We compare the distributions of $\Delta_\ell$ across depth groups using the Kruskal–Wallis test, with per-group means and positive-gain counts reported in Table 17. For ViT-based models, we group transformer blocks into Early (0–3), Middle (4–7), and Late (8–11), excluding non-block layers. For ResNet, we group layers by encoder stage: Early (stages 0–1), Middle (stage 2), and Late (stage 3), excluding non-stage layers (e.g., embedder/classifier).

For ViT-based models, Kruskal–Wallis on $\Delta_\ell$ yields $H(2) = 10.849$, $p = 4.41 \times 10^{-3}$. For ResNet, it yields $H(2) = 16.934$, $p = 2.10 \times 10^{-4}$.

## F. Overview of Neuron Interpretation

**Directional notation as message passing.** Consider a linear layer $\mathbf{y} = \mathbf{X}\mathbf{x}$, where $\mathbf{x} \in \mathbb{R}^n$ denotes input activations (input neurons) and $\mathbf{y} \in \mathbb{R}^m$ denotes output activations (output neurons), with weights $\mathbf{X} \in \mathbb{R}^{m \times n}$. We view $\mathbf{X}$ as a directed bipartite message-passing operator from inputs to outputs: each edge $(j \to i)$ carries the message $x_j$ scaled by weight $X_{i,j}$.

To make this directionality explicit, we define two complementary *weight patterns*:

$$\mathbf{X}_{:\to i} := \mathbf{X}_{i,:} \in \mathbb{R}^n, \tag{31}$$

$$\mathbf{X}_{j\to:} := \mathbf{X}_{:,j} \in \mathbb{R}^m. \tag{32}$$

Here, $\mathbf{X}_{:\to i}$ denotes the incoming weight pattern to output neuron $i$, while $\mathbf{X}_{j\to:}$ denotes the outgoing weight pattern from input neuron $j$.

**First-order probing as measuring alignment to learned directions.** Let $\mathbf{U} = [\mathbf{u}_1, \ldots, \mathbf{u}_r] \in \mathbb{R}^{n \times r}$ be learnable probe directions in the input space. The row probing branch

$$\mathbf{S}^{\mathrm{row}} = \mathbf{X}\mathbf{U} \in \mathbb{R}^{m \times r} \tag{33}$$

can be read neuron-wise as

$$\mathbf{S}^{\mathrm{row}}_{i,k} = \langle \mathbf{X}_{:\to i}, \mathbf{u}_k \rangle, \tag{34}$$

i.e., the $k$-th response at output neuron $i$ is the inner product between its incoming weight pattern and probe direction $\mathbf{u}_k$.

Thus, $\mathbf{X}\mathbf{U}$ summarizes each output neuron by how strongly it aligns with $r$ learned input-space landmarks.

Similarly, with $\mathbf{V} = [\mathbf{v}_1, \ldots, \mathbf{v}_r] \in \mathbb{R}^{m \times r}$, the column probing branch

$$\mathbf{S}^{\mathrm{col}} = \mathbf{X}^\top \mathbf{V} \in \mathbb{R}^{n \times r} \tag{35}$$

admits the interpretation

$$\mathbf{S}^{\mathrm{col}}_{j,k} = \langle \mathbf{X}_{j\to:}, \mathbf{v}_k \rangle, \tag{36}$$

so each input neuron/feature $j$ is summarized by how its outgoing weight pattern aligns with $r$ learned output-space landmarks.

**Gram matrices as intrinsic similarity maps over neurons.** Beyond alignment to external probe directions, the weight matrix itself induces *intrinsic similarity graphs*.

**Row (output-neuron) similarity.** The row Gram matrix

$$\mathbf{K}_{\mathrm{row}} = \mathbf{X}\mathbf{X}^\top \in \mathbb{R}^{m \times m} \tag{37}$$

has entries

$$(\mathbf{K}_{\mathrm{row}})_{i,j} = \langle \mathbf{X}_{:\to i}, \mathbf{X}_{:\to j} \rangle, \tag{38}$$

measuring how similar two output neurons are in terms of *which inputs they emphasize*. Large values indicate that neurons $i$ and $j$ place weight on similar input directions, forming a correlation structure among output neurons.

**Column (input-feature) similarity.** The column Gram matrix

$$\mathbf{K}_{\mathrm{col}} = \mathbf{X}^\top \mathbf{X} \in \mathbb{R}^{n \times n} \tag{39}$$

has entries

$$(\mathbf{K}_{\mathrm{col}})_{p,q} = \langle \mathbf{X}_{p\to:}, \mathbf{X}_{q\to:} \rangle, \tag{40}$$

measuring how similar two input features are in terms of *how they affect all outputs*. This yields a correlation structure among input features.

**Second-order probing as message passing on similarity graphs.** The second-order branches perform message passing not on the original bipartite input→output graph, but on the induced similarity graphs defined by $\mathbf{K}_{\mathrm{row}}$ and $\mathbf{K}_{\mathrm{col}}$.

**Row-kernel probing.** With $\mathbf{W} = [\mathbf{w}_1, \ldots, \mathbf{w}_r] \in \mathbb{R}^{m \times r}$,

$$\mathbf{S}^{\mathrm{row\text{-}kernel}} = \mathbf{K}_{\mathrm{row}}\mathbf{W} = \mathbf{X}\mathbf{X}^\top\mathbf{W} \in \mathbb{R}^{m \times r}. \tag{41}$$

For each probe $k$ and output neuron $i$,

$$(\mathbf{S}^{\mathrm{row\text{-}kernel}})_{i,k} = \sum_{j=1}^{m} w_{j,k} \langle \mathbf{X}_{:\to i}, \mathbf{X}_{:\to j} \rangle. \tag{42}$$

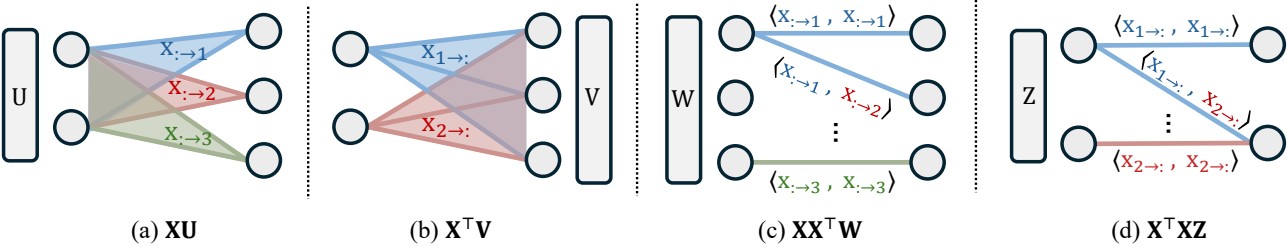

*Figure 5.* **Neuron-level interpretation of a weight matrix.** Illustration for $\mathbf{X} \in \mathbb{R}^{3 \times 2}$ (two input neurons, three output neurons). (a) $\mathbf{XU}$ corresponds to responses at each output neuron $i$, driven by its incoming weight pattern $\mathbf{X}_{:\to i}$. (b) $\mathbf{X}^{\top}\mathbf{V}$ corresponds to responses at each input neuron $j$, driven by its outgoing weight pattern $\mathbf{X}_{j\to:}$. (c) $\mathbf{XX}^{\top}\mathbf{W}$ probes the row-similarity structure $\langle \mathbf{X}_{:\to i}, \mathbf{X}_{:\to j} \rangle$. (d) $\mathbf{X}^{\top}\mathbf{XZ}$ probes the column-similarity structure $\langle \mathbf{X}_{p\to:}, \mathbf{X}_{q\to:} \rangle$.

This can be viewed as: neuron $i$ receives messages from other output neurons $j$, weighted by (i) their similarity $\langle \mathbf{X}_{:\to i}, \mathbf{X}_{:\to j} \rangle$ and (ii) landmark coefficients $w_{j,k}$. Hence, each column $\mathbf{w}_k$ selects a *landmark mixture* of output neurons, and the response summarizes how similar neuron $i$ is to that mixture.

**Column-kernel probing.** With $\mathbf{Z} = [\mathbf{z}_1, \ldots, \mathbf{z}_r] \in \mathbb{R}^{n \times r}$,

$$\mathbf{S}^{\text{col-kernel}} = \mathbf{K}_{\text{col}}\mathbf{Z} = \mathbf{X}^{\top}\mathbf{XZ} \in \mathbb{R}^{n \times r}, \qquad (43)$$

and for each input feature $p$,

$$(\mathbf{S}^{\text{col-kernel}})_{p,k} = \sum_{q=1}^{n} z_{q,k} \langle \mathbf{X}_{p\to:}, \mathbf{X}_{q\to:} \rangle. \qquad (44)$$

Thus, feature $p$ aggregates similarity-based messages from other features $q$ on the input-feature similarity graph.

First-order branches ($\mathbf{XU}$, $\mathbf{X}^{\top}\mathbf{V}$) quantify alignment of incoming/outgoing weight patterns to learned directions. Second-order branches ($\mathbf{XX}^{\top}\mathbf{W}$, $\mathbf{X}^{\top}\mathbf{XZ}$) additionally encode the *intrinsic correlation topology* among neurons/features by propagating over Gram-induced similarity graphs. This is precisely the intuition illustrated in Figure 5.

## G. Algorithm Pseudocode

Algorithm 1 summarizes the complete MVProbe forward pass, from the four probing branches through per-branch standardization, branch-specific projection, fusion, and classification. For completeness, Table 18 lists the weight-matrix dimensions of the best-performing layer used for each architecture.

*Table 18.* **Matrix dimensions ($\mathbf{X}^{m \times n}$) of the best-performing layer for each architecture.** Specifically, we use $l = 67, 59, 64, 47$ for ResNet, SupViT, MAE, DINO and $l = 46$ for $SD_{200}$ and $SD_{1k}$.

| ResNet | SupViT | MAE | DINO | SD$_{200}$ & SD$_{1k}$ |
|---|---|---|---|---|
| $1024 \times 256$ | $768 \times 768$ | $768 \times 768$ | $768 \times 3072$ | $1280 \times 1280$ |

---

**Algorithm 1** MVProbe: Multi-View Probing for Weight-Space Learning

**Require:** Weight matrix $\mathbf{X} \in \mathbb{R}^{m \times n}$, probe matrices $\mathbf{U}, \mathbf{Z} \in \mathbb{R}^{n \times r}$, $\mathbf{V}, \mathbf{W} \in \mathbb{R}^{m \times r}$, branch projections $\{\pi_i\}_{i=1}^{4}$, shared encoder $\psi$, classifier $\phi$
**Ensure:** Predicted label vector $\hat{\mathbf{y}} \in \mathbb{R}^{C}$
1: **// First-order branches**
2: $\mathbf{S}^{\text{row}} \leftarrow \mathbf{XU} \in \mathbb{R}^{m \times r}$      ▷ Row probing
3: $\mathbf{S}^{\text{col}} \leftarrow \mathbf{X}^{\top}\mathbf{V} \in \mathbb{R}^{n \times r}$      ▷ Column probing
4: **// Second-order branches (efficient computation)**
5: $\mathbf{S}^{\text{row-kernel}} \leftarrow \mathbf{X}(\mathbf{X}^{\top}\mathbf{W}) \in \mathbb{R}^{m \times r}$      ▷ Row kernel probing
6: $\mathbf{S}^{\text{col-kernel}} \leftarrow \mathbf{X}^{\top}(\mathbf{XZ}) \in \mathbb{R}^{n \times r}$      ▷ Column kernel probing
7: **// Per-sample standardization for each branch**
8: **for** $\mathbf{S} \in \{\mathbf{S}^{\text{row}}, \mathbf{S}^{\text{col}}, \mathbf{S}^{\text{row-kernel}}, \mathbf{S}^{\text{col-kernel}}\}$ **do**
9:      $\tilde{\mathbf{S}} \leftarrow \dfrac{\mathbf{S} - \mu(\mathbf{S})}{\sigma(\mathbf{S}) + \epsilon}$
10: **end for**
11: **// Branch-specific projection**
12: **for** $i \in \{1, 2, 3, 4\}$ **do**
13:      $\mathbf{f}_i \leftarrow \text{ReLU}(\pi_i(\tilde{\mathbf{S}}^{(i)})) \in \mathbb{R}^{d}$
14: **end for**
15: **// Feature fusion and classification**
16: $\mathbf{f} \leftarrow [\mathbf{f}_1; \mathbf{f}_2; \mathbf{f}_3; \mathbf{f}_4] \in \mathbb{R}^{4d}$
17: $\hat{\mathbf{y}} \leftarrow \phi(\psi(\mathbf{f})) \in \mathbb{R}^{C}$
18: **// Training objective**
19: $\mathcal{L} \leftarrow \mathcal{L}_{\text{BCE}}(\hat{\mathbf{y}}, \mathbf{y})$
20: **return** $\hat{\mathbf{y}}$

## H. Empirical Relevance of Theorem 4.1

Beyond the exact-equality construction in Theorem 4.1, near-collisions also matter empirically. We measure two quantities on *failure layers*—layers where ProbeX accuracy drops below 57% on Model Jungle or below 26% on Stable Diffusion: (i) the **pairwise overlap** between cosine-similarity distributions of positive and negative pairs, where a lower overlap coefficient (OVL) indicates better separation, and (ii) the **recovery rate** on samples that the row-side first-order probe $\mathbf{XU}$ alone fails on. Tables 19–22 together show that adding second-order branches sharply reduces representation collisions and rescues failure cases that single-view first-order probing misses.

**Notation.** Let $g$ be a branch representation (e.g., $\mathbf{XU}$, second-order, or full MVProbe); $\mathrm{NN}_k^g(x)$: $k$-th cosine-similarity neighbor under $g$; $J(a, b)$: label-set Jaccard similarity on Model Jungle. $\mathrm{OVL} = \int \min(f_+, f_-)\, ds$ (Inman & Bradley Jr, 1989); positive pairs: $J \geq 0.5$ on Model Jungle and same class on Stable Diffusion; negative pairs: $J \leq 0.25$ on Model Jungle and different class on Stable Diffusion. AUROC: area under the receiver operating characteristic for positive-versus-negative classification. The rescue rate on Model Jungle is the fraction of bottom-20% $\mathbf{XU}$-Jaccard samples whose $\mathrm{NN}_1^{g_t}$ Jaccard improves by at least $0.05$; the top-5 rescue rate on Stable Diffusion is the fraction of $\mathbf{XU}$-misclassified samples whose top-5 $\mathrm{NN}^{g_t}$ contains the correct class.

*Table 19.* **Pairwise embedding overlap on Model Jungle failure layers.** We report the overlap coefficient (OVL; lower is better) and AUROC (higher is better) between cosine-similarity distributions of positive and negative pairs, evaluated on layers where ProbeX accuracy drops below 57%. Best results are in **bold**.

| Layer | | $\mathbf{XU}$ | 1st | 2nd | Full |
|---|---|---|---|---|---|
| ResNet L57 | OVL ↓ | 0.014 | 0.005 | 0.112 | **0.000** |
| | AUC ↑ | 0.989 | 0.997 | 0.786 | **0.998** |
| SupViT L66 | OVL ↓ | **0.003** | 0.155 | 0.060 | 0.060 |
| | AUC ↑ | **0.998** | 0.778 | 0.921 | 0.959 |
| DINO L49 | OVL ↓ | 0.761 | 0.639 | **0.084** | 0.578 |
| | AUC ↑ | 0.559 | 0.630 | **0.758** | 0.639 |
| MAE L42 | OVL ↓ | 0.192 | 0.148 | 0.195 | **0.146** |
| | AUC ↑ | 0.726 | **0.793** | 0.493 | 0.671 |

## I. Efficiency Analysis

We profile training time, throughput, and peak GPU memory across all evaluated weight matrix shapes (Table 23). Implementation uses the official ProbeX codebase with standard PyTorch operations only; no custom CUDA kernels are used.

*Table 20.* **Pairwise embedding overlap on SD LoRA failure layers.** OVL (lower is better) and AUROC (higher is better) between same-class and different-class cosine-similarity distributions, evaluated on layers where ProbeX accuracy drops below 26%. Best results are in **bold**.

| Setting | | $\mathbf{XU}$ | 2nd | Full | $\mathbf{XX}^\top\mathbf{U}$ | $\mathbf{X}^\top\mathbf{XU}$ |
|---|---|---|---|---|---|---|
| SD_200 L110 (In-Dist) | OVL ↓ | 0.511 | 0.108 | 0.126 | **0.086** | 0.904 |
| | AUC ↑ | 0.816 | 0.988 | 0.979 | **0.989** | 0.500 |
| SD_200 L110 (Zero-shot) | OVL ↓ | 0.587 | **0.305** | 0.309 | 0.325 | 0.951 |
| | AUC ↑ | 0.771 | **0.926** | 0.923 | 0.917 | 0.498 |
| SD_1k L94 (In-Dist) | OVL ↓ | 0.378 | 0.058 | **0.036** | 0.052 | 0.814 |
| | AUC ↑ | 0.855 | 0.994 | **0.996** | 0.995 | 0.507 |
| SD_1k L94 (Zero-shot) | OVL ↓ | 0.469 | 0.095 | **0.086** | 0.095 | 0.919 |
| | AUC ↑ | 0.846 | 0.987 | **0.989** | 0.987 | 0.501 |

*Table 21.* **Local-neighborhood recovery on Model Jungle.** Rescue rate (%) on the bottom-20% of samples by $\mathbf{XU}$-Jaccard similarity. $|\mathcal{F}_s|$ denotes the source-hard subset size.

| Layer | $|\mathcal{F}_s|$ | $\mathbf{XU}\to$2nd | $\mathbf{XU}\to$Full |
|---|---|---|---|
| ResNet L57 | 54 | 22.2 | 51.9 |
| SupViT L66 | 59 | 33.9 | 32.2 |
| DINO L49 | 43 | 53.5 | 46.5 |
| MAE L42 | 45 | 53.3 | 53.3 |

*Table 22.* **Local-neighborhood recovery on SD LoRA.** Top-5 rescue rate (%) on samples that $\mathbf{XU}$ top-1 misclassifies. $|\mathcal{F}_s|$ denotes the source-hard subset size.

| Setting | $|\mathcal{F}_s|$ | $\mathbf{XU}\to$2nd | $\mathbf{XU}\to$Full |
|---|---|---|---|
| SD_200 L110 (In-Dist) | 472 | 89.0 | 87.1 |
| SD_200 L110 (Zero-shot) | 429 | 94.2 | 90.7 |
| SD_1k L94 (In-Dist) | 491 | 35.0 | 35.4 |
| SD_1k L94 (Zero-shot) | 438 | 92.9 | 93.4 |

*Table 23.* **Runtime and memory comparison on RTX 3090 (24 GiB).** We report wall-clock training time per epoch (s/epoch), throughput (samples/s), and peak GPU memory (MiB) across all evaluated weight matrix shapes. ViT avg denotes the mean over SupViT, MAE, and DINO (all $768\times768$).

| Family | Method | Shape ($m\times n$) | s/epoch | Throughput | Peak Mem |
|---|---|---|---|---|---|
| ResNet | ProbeX | $1024\times256$ | 1.19 | 593 | 412 |
| | PX(×4) | $1024\times256$ | 1.28 | 550 | 1205 |
| | MVProbe | $1024\times256$ | 1.31 | 535 | 1513 |
| ViT avg | ProbeX | $768\times768$ | 1.86 | 375 | 540 |
| | PX(×4) | $768\times768$ | 1.88 | 369 | 1239 |
| | MVProbe | $768\times768$ | 2.05 | 339 | 2172 |
| SD_200 | ProbeX | $1280\times1280$ | 1.74 | 2009 | 2676 |
| | PX(×4) | $1280\times1280$ | 1.85 | 1895 | 3069 |
| | MVProbe | $1280\times1280$ | 2.58 | 1358 | 5152 |
| SD_1k | ProbeX | $1280\times1280$ | 2.29 | 1566 | 2679 |
| | PX(×4) | $1280\times1280$ | 2.50 | 1400 | 3072 |
| | MVProbe | $1280\times1280$ | 2.87 | 1220 | 5156 |

## J. Fusion Strategy Sensitivity

We test whether MVProbe's gain hinges on a particular fusion choice. We compare four configurations: per-sample standardization or per-branch $L_2$ normalization, each combined with simple concatenation, learned group weighting between the first- and second-order groups, or learned per-branch weighting. As shown in Table 24, per-branch $L_2$ normalization is clearly insufficient (mean accuracy 73.40), and learned adaptive weighting never outperforms simple concatenation while being unstable across architecture families (the group weights swing from 0.876/0.124 on ResNet to 0.562/0.438 on DINO). The key design issue is pre-fusion branch conditioning (Theorem 4.3, Corollary 4.4), not fusion complexity.

*Table 24.* **Fusion sensitivity across architectures** (accuracy %). **Bold** = best per column.

| Condition | ResNet | DINO | SupViT | MAE | Avg |
|---|---|---|---|---|---|
| Std + Concat (Ours) | **91.79** | **78.31** | 91.90 | **81.47** | **85.87** |
| Std + Group Weight | 91.47 | 78.14 | **92.33** | 80.40 | 85.59 |
| Std + Branch Weight | 90.25 | 77.47 | 91.02 | 79.89 | 84.66 |
| L2 + Concat | 83.47 | 62.59 | 75.48 | 72.06 | 73.40 |

## K. SD LoRA Failure-Layer Analysis

We isolate *failure layers*—SD LoRA layers where ProbeX accuracy drops below 26%—to test whether second-order branches are decisive precisely where first-order probing breaks down. As shown in Table 25, all first-order variants collapse to below 21% on these layers, while the full MVProbe maintains 62–90% accuracy. This confirms that the multi-view combination is essential rather than redundant, and that the second-order branches recover information that first-order probing cannot represent in this regime.

*Table 25.* **Classification accuracy (%) on SD LoRA failure layers** (ProbeX accuracy below 26%). The first-order column reports the combined first-order pair $\mathbf{X}\mathbf{U}$ and $\mathbf{X}^\top\mathbf{V}$; the full column reports the complete four-branch MVProbe.

| Setting | ProbeX | First-order | Full |
|---|---|---|---|
| SD_200 L110 (In-Dist) | 16.0 | 13.6 | **88.8** |
| SD_200 L110 (Zero-shot) | 23.6 | 18.6 | **61.7** |
| SD_1k L94 (In-Dist) | 9.4 | 9.8 | **89.4** |
| SD_1k L94 (Zero-shot) | 25.2 | 20.6 | **89.8** |

## L. Comparison on the Small-scale benchmark

We provide an additional comparison against equivariant weight-space methods (Zhang et al., 2023; Navon et al., 2023; Kofinas et al., 2024; Lim et al., 2024) on the small-scale MNIST INR benchmark, where INR checkpoints trained on MNIST serve as input for predicting the corresponding labels. As MVProbe targets single-layer probing

in large-scale settings, we adapt it to this benchmark, where probing all layers is feasible given the small weight matrices. The INR comprises three layers—$F := \mathbf{X}_3 \circ \sigma \circ \mathbf{X}_2 \circ \sigma \circ \mathbf{X}_1$, where $\sigma$ is the activation—and we compute row probing as $F(\mathbf{U})$, column probing as $\mathbf{X}_i^\top\mathbf{V}$ at the best-performing layer, and second-order probing as $\mathbf{X}_i\mathbf{X}_i^\top\mathbf{W}$ and $\mathbf{X}_i^\top\mathbf{X}_i\mathbf{Z}$. For this benchmark, we set the projection dimension to $d=8$, the number of probes per branch to $r=128$, and the hidden dimension to 128. As shown in Table 26, MVProbe achieves the best performance. These results show that multi-view probing transfers gracefully beyond its intended large-scale, single-layer regime, supporting its use as a general-purpose probing framework.

*Table 26.* Comparison on MNIST INR. The best result in each column is in **bold** and the second best is underlined. All training/inference times are measured on a single NVIDIA RTX 3090 GPU.

| Method | Acc. | Train. (s/ep) | Infer. (s) | # Params |
|---|---|---|---|---|
| Zhang et al. (Zhang et al., 2023) | 96.28 | 395.60 | 0.99 | 484,298 |
| Navon et al. (Navon et al., 2023) | 86.50 | 656.09 | 3.30 | 552,650 |
| Kofinas et al. (Kofinas et al., 2024) | 89.37 | 316.69 | 1.03 | **376,586** |
| Lim et al. (Lim et al., 2024) | 90.23 | **95.00** | **0.44** | 716,426 |
| MVProbe (Ours) | **97.20** | 272.60 | 0.50 | 605,716 |

