# OpenReview forum: "What Linear Probes Miss: Multi-View Probing for Weight-Space Learning"
_ICML.cc/2026/Conference — ICML 2026 regular_

### Official Review · Reviewer_XDVk · 2026-03-01

**Soundness:** 3
**Presentation:** 3
**Significance:** 2
**Originality:** 2
**Overall Recommendation:** 4
**Confidence:** 3

**Summary:**

This paper studies weight space learning. Existing probe methods are mostly single-view and capture mainly first-order structure, missing richer higher-order row/column interaction patterns. They introduce MVProbe, a multi-view probing framework that combines first-order signals with interaction-aware, Gram-based views. These multi-perspective representations are shown to be more effective with empirical experiments and more expressive through theoretical evidence.

**Compliance With Llm Reviewing Policy:**

Affirmed.

**Final Justification:**

The new stable diffusion experiment is promising, and i decide to improve my score.

**Key Questions For Authors:**

1. In Table 2, XU is already much better than all baselines. Is this a typo? Based on this result, multi-view is not that helpful.
2. What's the scale of the problem, like the size of these matrix dimensions? For large-scale cases, i think you need some more complex batching designs to handle these Gram matrices.
3. Based on Table 5, do the implementations of ProbeX and this method share similar code-level optimization? It seems ProbeX has much less computational complexity, while the running time is nearly the same.
4. Can these methods scale to larger models like LLMs? That would be valuable.

**Limitations:**

Limitations are discussed, but I don't think they're the key problem. My biggest concern is that most studies are still based on some "toy" cases (with each training epoch around 1s). Can these studies provide real insights for large-scale models?

**Strengths And Weaknesses:**

Strengths:
1. claim is supported by theoretical evidence
2. motivation is very clear, first order is not expressive -> introducing higher-order interaction
3. as shown in Figure3, these higher-order interactions reduce cross-layer variance, which is a good property


Weakness:
1. the first paragraph in the introduction is a bit misleading. I would assume this paper introduces something which can be used to deduce the properties of real-world models like LLMs, while the experiment section is more about some "toy" experiments
2. The theoretical part just shows some obvious conclusions, providing limited insights, especially the first expressive results
3. in table 5, this method shows that it requires substantially more computation. And I guess the training time is based on some code-level optimization.
4. no implementation details are given and no code provided.
5. I am confused about some results, please see question section

---

> ### Author Rebuttal · Authors · 2026-03-31
>
> For reproducibility (including rebuttal results), we share our anonymous code: https://anonymous.4open.science/r/mv1
>
> **Q1 (Table 2 — XU already better than baselines?):**
> Not a typo. We agree that Tab.2 may give the impression that XU only is sufficient. However, our main takeaway is that multi-view remains helpful. The gap between XU only and prior baselines is influenced by differences in our setting, including per-sample standardization and backbone choice.
>
> Tab.2 is a within-framework ablation illustrating complementarity, rather than as direct comparison against prior baselines. Even within the 1st-order, adding the column view improves over XU consistently. With all four branches: +1.82 (ResNet), +3.24 (SupViT), +3.27 (MAE), +4.12 (DINO) over XU. This shows that different operators capture complementary structural signals, and that combining them provides more complete characterization than any single branch alone. To avoid this confusion, we will revise the caption in Tab.2 to make this clearer.
>
> **Q2 (Matrix dimension scale):**
> We first clarify the matrix scale, and then explain why 2nd-order branches do not cause computational blow-up. In our experiments, the probed weight matrices have sizes 1024×256 (ResNet), 768×768 (SupViT/MAE/DINO), and up to 1280×1280 (Stable Diffusion LoRA; newly added, dataset released Feb. 11), with r=128 probing vectors.
>
> MVProbe never forms Gram matrices explicitly. It uses associative evaluation: X(XᵀW) instead of (XXᵀ)W. This costs O(mnr) — the same order as 1st-order probing — versus O(m²n+m²r) for the naive form. Please see Q3+W3+W4 below for further discussions.
>
> **Q3+W3+W4 (Code-level optimization / Running time):**
> MVProbe is built directly on the official ProbeX codebase, without code-level optimizations.
>
> The core change is simply computing complementary projections and concatenating [(Link: pseudocode)]( https://anonymous.4open.science/r/mv1/a/f6.png). Thus, the difference is only a few additional projection lines while the rest of the pipeline remains unchanged. The similar runtime despite higher FLOPs is expected because these projections are independent matrix multiplications that are efficiently parallelized on modern GPUs. As a result, although MVProbe has higher FLOPs, training time per epoch remains very close to ProbeX in practice.
>
> See the full [(Link: profiling)](https://anonymous.4open.science/r/mv1/a/f7.png).
>
> **Q4 (Scale to larger models?):**
> Yes, MVProbe operates directly on individual weight matrices regardless of architecture or training objective. Since LLMs also consist of linear weight matrices (e.g., attention and MLP projections), the same probing operators can be applied.
>
> We have evaluated MVProbe on foundation-model-level benchmark (Stable Diffusion LoRA models), achieving 97.88% vs ProbeX 35.75% on SD_1k. Please see our response to LtVP (Q1+W1) for full results. Exploring probing at full LLM scale is an exciting direction for future work.
>
> **W1+L1 (Misleading intro / "toy" experiments):**
> The experiments may appear small without context, but Model Jungle is currently the most comprehensive benchmark in weight-space learning. Earlier probing works (e.g., ProbeGen, NG-T) were evaluated on much smaller settings such as 3-layer INR or shallow CNNs, whereas Model Jungle contains modern architectures including ResNet101 and ViT-B with 60+ layers.
> The short training time (~1–3s/epoch) reflects the efficiency of single-layer probing rather than the simplicity of the problem. Specifically, we extract single-layer matrix from full-scale networks (up to 1280×1280) and classify model categories.
>
> We will revise the introduction to better align expectations and clarify that this work focuses on improving probing operators for weight-space learning, with extensions to larger systems (e.g., LLMs) as future work.
>
> **W2 (Theoretical results seem obvious):**
> Our theoretical contribution identifies and resolves consequential bottlenecks that have been overlooked in weight-space learning. Despite its conceptual simplicity, Thm. 4.1 formalizes a practically important limitation of standard first-order probing: structurally distinct weight matrices can produce identical 1st-order outcomes. This is not merely a theoretical corner case; it becomes particularly consequential in poorly informative layers, where ProbeX collapses (Fig. 3)—precisely where MVProbe achieves pronounced improvements.
>
> Thm. 4.2 addresses a second practical issue: scale imbalance b.w. 1st- and 2nd-order branches, which we analyze using a rotational symmetry argument. Cor. 4.3 shows that standardization resolves this imbalance, making it a principled and essential part of our method.
>
> Together, these results expose previously overlooked limitations of standard probing methods and motivate the multi-view design of MVProbe.

---

> > ### Author Rebuttal · Reviewer_XDVk · 2026-04-02
> >
> > Thanks for the response. The newly added experiment seems promising, and i will raise my score to 4.

---

> > > ### Author Response · Authors · 2026-04-04
> > >
> > > Thank you for your thoughtful review and for raising your score. Regarding the reviewer's theoretical concern, we have also added a theoretical discussion in our response to Reviewer HkVt that may be of interest: in the pairwise distinguishability sense, any weight pair distinguished by ProbeX (×4) can also be distinguished by MVProbe. We further discuss the stronger fixed-budget comparison in the same response.

---

### Official Review · Reviewer_LtVP · 2026-03-08

**Soundness:** 3
**Presentation:** 4
**Significance:** 3
**Originality:** 2
**Overall Recommendation:** 4
**Confidence:** 3

**Summary:**

This paper studies the problem of weight-space learning, where the goal is to infer properties of neural networks directly from their parameters. The authors proposed MVProve, which consists of a multi-branch probing framework that extracts information from the weight matrices using a combination of first-order and second-order probing.

**Compliance With Llm Reviewing Policy:**

Affirmed.

**Final Justification:**

The authors have not fully resolved the issues of the paper, as detailed in my reply to the rebuttal.

**Key Questions For Authors:**

- Have the authors evaluated the method on other weight-space learning tasks beyond predicting training label sets?

- Are there cases where the additional branches introduce redundancy or negatively affect performance (especially the second-order one)?

**Limitations:**

Yes

**Strengths And Weaknesses:**

Strengths

- Timely problem: Weight-space learning is an increasingly important research direction given the rapid growth of publicly available model checkpoints. Hence, frameworks proposed in this space could be useful in settings where metadata is unavailable or incomplete

- Presentation: The paper is well-structured and easy to follow.

- Theoretical component: The reviewer appreciates the theoretical derivations found in the paper.

Weaknesses:

- Pure focus on Model jungle dataset: The empirical evaluation focuses primarily on the Model Jungle dataset and the task of predicting training categories from weight matrices. It is unclear how the proposed method could perform in other weight learning tasks, such as identifying architectural characteristics.

- Complexity: The multi-branch architecture introduces additional projections and processing steps compared to simpler probing approaches (especially given the second-order aspect). A more detailed comparison of efficiency and scalability would add value to the paper.

- Performance: It seems that first-order probing (table 2) provides a very good baseline of performance and the second order aspect of the proposed methodology provides marginal improvement. It would be interesting to see in which cases such improvement comes and in what cases does the additional second order component bring performance value.

---

> ### Author Rebuttal · Authors · 2026-03-31
>
> For reproducibility (including rebuttal results), we share our anonymous code: https://anonymous.4open.science/r/mv1
>
> **Q1+W1 (Scalability to larger benchmarks and other tasks):**
>
> **1. Scalability**
> We evaluated MVProbe on **previously unseen benchmark setting released after submission**: Stable Diffusion LoRA models, involving generative foundation models rather than discriminative networks.
>
> SD LoRA Classification (Layer 46, 5 seeds):
> |Method|Params|SD_200 (In-Dist)|SD_200 (Zero-shot)|SD_1k (In-Dist)|SD_1k (Zero-shot)|
> |:-|:-:|:-:|:-:|:-:|:-:|
> |ProbeX|9.1M|98.48±0.48|94.01±0.77|35.75±2.44|52.42±2.48|
> |ProbeX(×4)|34.8M|97.72±0.50|93.53±1.99|32.46±3.08|51.14±3.88|
> |**MVProbe (Ours)**|35.3M|**99.80±0.00**|**95.53±0.65**|**97.88±0.37**|**97.96±0.29**|
>
> **Note)** In-Dist: On the same data distribution as training set; Zero-shot: On non-overlapped distributions.
>
> MVProbe outperforms ProbeX and (x4), achieving near-perfect performance on SD_1k. This indicates that the gain arises from our multi-view representation design rather than capacity.
>
> This improvement is not confined to a single selected layer. As shown in the [(Link: layer-wise SD)](https://anonymous.4open.science/r/mv1/a/f4.png), MVProbe consistently outperforms ProbeX across layers, with margins up to +83.57% (SD_1k L70 In-Dist) and +72.85% (SD_1k L94 Zero-shot), confirming a regime-wide phenomenon.
>
> **2. Other tasks**
>
> Model Jungle is currently the most comprehensive benchmark for weight-space learning. Earlier probing works (e.g., ProbeGen, NG-T) were evaluated on much smaller settings (e.g., 3-layer INR, shallow CNNs), while Model Jungle includes modern architectures (e.g., ResNet101, ViT-B with 60+ layers). As a result, standardized benchmarks for other tasks remain limited.
>
> To test whether the learned representation generalizes beyond model classification, we evaluate MVProbe on kNN retrieval and OOD detection on Stable Diffusion weights. kNN retrieval labels a weight matrix by the majority class of its nearest neighbors, while OOD detection uses the AUC of kNN similarity scores to distinguish in- vs. out-of-distribution weights. Both tasks measure whether the learned representation forms coherent clusters in weight space.
>
> On SD_1k, MVProbe achieves 0.968 in kNN retrieval (k=1) compared to ProbeX (0.212), with similarly strong gains on OOD detection, indicating that the learned embedding is well-structured in weight space rather than tied to a classifier head.
>
> SD_200:
> |k|kNN (PX)|kNN (Ours)|OOD kNN (PX)|OOD kNN (Ours)|
> |:-:|:-:|:-:|:-:|:-:|
> |1|0.958|**0.998**|0.779|**1.000**|
> |3|0.926|**0.998**|0.787|**1.000**|
> |5|0.914|**0.998**|0.793|**1.000**|
> |10|0.874|**0.998**|0.802|**1.000**|
> |24†|0.804|**0.994**|0.511|**0.515**|
>
> SD_1k:
> |k|kNN (PX)|kNN (Ours)|OOD kNN (PX)|OOD kNN (Ours)|
> |:-:|:-:|:-:|:-:|:-:|
> |1|0.212|**0.968**|0.721|**1.000**|
> |2|0.162|**0.970**|0.763|**0.997**|
> |3|0.156|**0.960**|0.774|**0.963**|
> |4†|0.136|**0.946**|0.703|**0.793**|
>
> **Note)** PX: ProbeX; †= max samples per class minus 1 (SD\_200: 25 samples, SD\_1k: 5 samples).
>
> These results suggest that MVProbe improves observability of weight-space structure rather than adapting to a particular benchmark.
>
> **Q2 (Branch negative effect or redundancy?):**
>
> **1. Negative effect.** We do not observe a net negative effect from combining branches. Because each branch probes the weight matrix through a different operator family (row-side, column-side, and interaction-aware 2nd-order), combining views provides a more complete characterization of the weight space. In Tab.2, the full four-branch model (MVProbe) consistently achieves the best performance across all architectures, indicating complementary rather than harmful in combination.
>
> **2. Redundancy.** Increasing probes within the same 1st-order operator family in ProbeX(×4), it still underperforms MVProbe. In contrast, MVProbe introduces structurally distinct operators and achieves stronger results, which provide complementary information and lead to consistent gains.
>
> **W2 (Complexity/Efficiency):**
> Please see our response to XDVk (Q2, Q3). We will add further analysis in the revision.
>
> **W3 (1st-order strong, 2nd-order marginal?):**
> The role of the 2nd-order branches becomes especially clear at failure layers. In layer-wise comparisons (Fig.3), single 1st-order (ProbeX) has several failure layers with near-random prediction. However, adding the 2nd-order pair (MVProbe) yields consistent gains across architectures (+36.85 ResNet, +25.59 SupViT, +10.24 MAE, +14.97 DINO). Also, the same pattern appears even more strongly on [(Link: Ablation on SD)](https://anonymous.4open.science/r/mv1/a/f5.png), where failure layers with ~10–21% 1st-order accuracy are recovered to 62–90% by the full model.
>
> These indicate that the 2nd-order branches add value precisely when 1st-order probing cannot capture the relevant weight-space structure.

---

> > ### Author Rebuttal · Reviewer_LtVP · 2026-04-02
> >
> > Thank you for the clarifications. I am maintaining my positive assessment of the paper and my current score.

---

> > > ### Author Response · Authors · 2026-04-05
> > >
> > > Thank you again for your positive assessment of our paper and for taking the time to read our rebuttal.
> > >
> > > We also wanted to mention that we added a theoretical discussion in our response to Reviewer HkVt that may be of interest, showing that MVProbe is at least as expressive as ProbeX (×4). We sincerely appreciate your thoughtful feedback and are grateful that our clarifications were helpful.

---

### Official Review · Reviewer_EGua · 2026-03-13

**Soundness:** 3
**Presentation:** 3
**Significance:** 2
**Originality:** 2
**Overall Recommendation:** 3
**Confidence:** 3

**Summary:**

This paper studies the problem of identifying properties of neural networks directly from their weights in the weight-space learning setting. The authors focus on probing-based methods, which avoid processing full model parameters directly by instead learning probe vectors that produce permutation-equivariant responses from selected weight matrices. The paper argues that existing single-view linear probing methods are fundamentally limited: they are asymmetric between row and column structure and can miss higher-order interaction patterns encoded by pairwise correlations. To address this, the authors propose MVProbe, a four-branch multi-view probing framework that combines row-wise and column-wise first-order projections with Gram-based second-order probing from both perspectives. The method also includes a per-sample standardization step motivated by a scaling analysis showing that higher-order branches can otherwise dominate the combined representation. On the theory side, the paper shows that second-order probing can distinguish weight matrices that are indistinguishable under first-order probing alone, and provides a result explaining the need for feature standardization across branches. Empirically, MVProbe is evaluated on the Model Jungle benchmark across ResNet, SupViT, MAE, and DINO model families, where it consistently outperforms prior probing baselines and exhibits greater robustness across layer choices.

**Compliance With Llm Reviewing Policy:**

Affirmed.

**Final Justification:**

I appreciate the authors' detailed rebuttal, which satisfactorily addressed several of my main concerns and improved my overall view of the paper. In particular, the responses clarified the practical role of the multi-view combination, provided a more convincing explanation of why the column-side and second-order branches are not redundant, and added useful additional evidence on fusion design and transfer beyond the original benchmark. These clarifications make the paper's contribution appear more substantive than I initially assessed. While I still view the method as somewhat incremental in a broad sense, I am now persuaded that the paper identifies a real structural limitation of single-view first-order probing and addresses it in a technically sound and empirically meaningful way. Overall, the rebuttal resolved my main doubts, and I am satisfied with raising my score accordingly.

**Key Questions For Authors:**

1. The paper argues that single-view first-order probing is fundamentally limited, and that adding row/column views together with second-order probing is the key reason for the improvement of MVProbe. Could the authors clarify which part is the most essential in practice? For example, is the main gain coming from adding the missing row/column symmetry, from introducing Gram-based second-order information, or from the combination of both?

2. The theoretical results show that second-order probing can distinguish some cases that first-order probing cannot, which is useful motivation. However, could the authors clarify how representative these separating examples are for the actual benchmark setting? In particular, do the authors have either empirical or theoretical evidence that such first-order indistinguishability is a practically important bottleneck in Model Jungle, rather than a possibility that only occurs in specially constructed cases?

3. MVProbe introduces four branches together with per-sample standardization before fusion. Could the authors provide more insight into why this particular fusion design is the right one? For instance, did the authors consider adaptive weighting, learned branch fusion, or simpler normalization schemes, and if so, how sensitive are the results to these choices?

4. The experiments are all conducted on the Model Jungle benchmark and focus on model-family property prediction. Could the authors discuss how broadly they expect the method to transfer beyond this setting? For example, would the same probing strategy be expected to help on other weight-space tasks such as retrieval, clustering, or cross-architecture comparison, or is the method mainly tuned to the specific benchmark studied here?

5. The paper improves over prior probing baselines, but the overall methodological change seems somewhat incremental. Could the authors explain more explicitly what they view as the main conceptual advance of MVProbe relative to prior probing frameworks? In particular, what should readers take away beyond the message that adding additional views and higher-order features improves probing quality?

**Limitations:**

Yes.

**Strengths And Weaknesses:**

The paper is technically competent and clearly written. Its main strength is that it identifies a concrete limitation of existing probing-based methods for weight-space learning, namely their reliance on single-view first-order projections, and proposes a natural extension that combines row-wise and column-wise perspectives with both first-order and second-order probing. The method is easy to understand, the design is reasonably well motivated, and the paper provides a useful theoretical observation showing that second-order probing can distinguish cases that first-order probing cannot. The empirical evaluation is also fairly comprehensive within the chosen benchmark, covering several model families and including ablations that support the contribution of the different branches and the standardization step. Overall, I found the work sound and the presentation clear.

My main reservations concern significance and originality. While the method is sensible and appears to improve over prior probing baselines, the contribution feels relatively specialized, in that it mainly strengthens a probing module within an existing weight-space learning pipeline rather than opening up a broader new direction. In addition, the proposed multi-view and second-order construction seems more like a natural extension of existing probing ideas than a fundamentally new conceptual advance. The theoretical results are also somewhat limited in scope and mainly serve to justify the design intuition, rather than to provide a deeper understanding of probing in weight space.

---

> ### Author Rebuttal · Authors · 2026-03-31
>
> For reproducibility (including rebuttal results), we share our anonymous code: https://anonymous.4open.science/r/mv1
>
> **Q1 (Which part is most essential?):**
> The essential contribution is the multi-view combination itself, supported by our ablation (Tab.2). 1st-order probing has structural blind spots in complementary column info. and row-column interaction structure. Neither the 1st-order pair nor the 2nd-order pair alone matches the full MVProbe across architectures, indicating that the gain arises from combining complementary operator views rather than any single component. Thus, MVProbe addresses a structural limitation of single-view 1st-order probing, an incomplete representation family for weight-space learning. For further details, please see our response to XDVk (Q1).
>
> **Q2 (Practical relevance of Thm 4.1):**
> Thm 4.1 is a structural existence result showing that 1st-order probing can be intrinsically unable to distinguish certain weight configurations. Beyond exact equality (X₁U = X₂U), the practical issue is near-collisions where distinct weight matrices produce similar 1st-order responses. We measure this using the overlap coefficient (OVL) between positive and negative cosine similarities (lower=better). Here, SD_200 and SD_1k denote two Stable Diffusion LoRA benchmark settings, and In-Dist and Zero-shot denote in-distribution and held-out evaluation, respectively:
>
> |Setting|XU OVL|2nd-order OVL|Reduction|
> |-|-|-|-|
> |DINO L49|0.761|**0.084**|9.1×|
> |SD_1k L94 (In-Dist)|0.378|**0.058**|6.5×|
> |SD_1k L94 (Zero-shot)|0.469|**0.095**|4.9×|
>
> We also analyze recovery on layers where 1st-order probing fails:
>
> |Setting|XU→2nd|XU→Full|
> |-|-|-|
> |SD_200 L110 (In-Dist)|89.0%|87.1%|
> |SD_200 L110 (Zero-shot)|94.2%|90.7%|
> |SD_1k L94 (In-Dist)|35.0%|35.4%|
> |SD_1k L94 (Zero-shot)|92.9%|93.4%|
>
> This matches the practical regime: MVProbe outperforms ProbeX on 95.1% of all layers (App. Tab.9), with the largest gains appearing where ProbeX performance degrades, indicating that the theoretical blind spots correspond to practically relevant limitations. Please see [(Link: OVL & recovery)](https://anonymous.4open.science/r/mv1/a/f2.png) for details.
>
> **Q3 (Fusion design choice):**
> We tested 3 more fusion strategies: (1) per-branch L2-norm+concat, (2) learned group weighting, (3) learned per-branch weighting. Please see [(Link: fusion EXP)](https://anonymous.4open.science/r/mv1/a/f3.png) for the detailed settings. The key design issue is pre-fusion branch conditioning (Thm 4.2, Cor 4.3), not complex fusion mechanisms.
>
> L2-norm+concat is clearly insufficient (avg 73.40 vs ours 85.87). Learned weighting — whether group-level (85.59) or per-branch (84.66) — never outperforms simple concat (85.87) and is unstable across datasets.
>
> **Q4 (Generalization beyond Model Jungle):**
>
> Our method is about single weight matrix, hence task- and benchmark-agnostic. We evaluate on a **previously unseen setting unavailable at submission: Stable Diffusion LoRA models (released Feb 11)**, as well as kNN retrieval and OOD detection. Please see our response to LtVP (Q1).
>
> **Q5 + W1 (Conceptual advance / Significance):**
>
> The key conceptual advance of MVProbe is not simply adding probe branches, but identifying a structural limitation of single-view 1st-order probing (e.g., XU). Our analysis shows that this representation is structurally incomplete: certain differences b.w. models cannot be detected under such operators.
>
> MVProbe addresses this limitation by introducing complementary operator families—cross-view 1st-order projections and interaction-aware 2nd-order operators—that expose aspects of the weight matrix invisible to single-view probing. From this perspective, the contribution is not merely increasing probe capacity, but clarifying what info. probing operators can or cannot capture in weight space.
> Our contributions are threefold:
>
> **(1) First formal characterization of probing limitations.** Thm 4.1 shows that 1st-order probing has intrinsic indistinguishability regions, providing principled explanations for why certain weight differences cannot be detected by prior methods.
>
> **(2) A principle for multi-order fusion.** The scaling analysis (Thm 4.2) and resulting standardization strategy are not specific to our four branches—they apply to any setting where features of different polynomial orders are combined. We believe this principle is reusable beyond MVProbe.
>
> **(3) Regime-dependent probing behavior.** Our experiments reveal, for the first time, that the informative probing order varies dramatically across architectures and layers (Fig.3), shifting the practical recommendation from "use XU" to combining complementary views—a qualitative shift in how practitioners should approach weight-space learning.
>
> Together, these results frame probing as a question of representation completeness in weight space rather than an empirical design choice, and MVProbe provides a principled basis for constructing probing representations.

---

> > ### Author Rebuttal · Reviewer_EGua · 2026-04-01
> >
> > Thank you for the clarification and additional studies. The rebuttal has satisfactorily addressed several of my main concerns, and I am willing to raise my score from 2 to 3.

---

> > > ### Author Response · Authors · 2026-04-05
> > >
> > > Thank you for taking the time to read our rebuttal and for updating your score.
> > >
> > > You mentioned that several of your main concerns were satisfactorily addressed, and we sincerely appreciate this update.
> > >
> > > If possible, it would be very helpful for us, as well as for the AC, to understand which specific concerns you feel remain unresolved and continue to support your current recommendation.
> > >
> > > In particular, our rebuttal addressed the following points raised in your original review:
> > >
> > > • the essential contribution and its relation to prior probing work
> > >
> > > • the practical relevance of Theorem 4.1
> > >
> > > • additional fusion experiments
> > >
> > > • evaluation beyond ResNet, SupViT, MAE, and DINO, including Stable Diffusion LoRA models
> > >
> > > Additionally, we have added a theoretical discussion in our response to Reviewer HkVt that may also be of interest, showing that MVProbe is at least as expressive as ProbeX (×4).
> > >
> > > If possible, could you indicate which of these aspects you still find insufficient?
> > >
> > > Thank you again for your time and thoughtful feedback.

---

### Official Review · Reviewer_HkVt · 2026-03-19

**Soundness:** 3
**Presentation:** 3
**Significance:** 3
**Originality:** 3
**Overall Recommendation:** 4
**Confidence:** 4

**Summary:**

This paper introduces **MVProbe**, a method for learning properties of neural networks directly from their weights. The key idea is to improve efficiency by probing only a single layer, while increasing expressivity compared to prior work (e.g., ProbeX) by incorporating second-order information through weight correlations in addition to standard first-order probes.

Empirically, the method demonstrates consistent improvements over existing scalable weight-space learning baselines, while maintaining comparable computational complexity.

**Compliance With Llm Reviewing Policy:**

Affirmed.

**Final Justification:**

Most of my comments on the initial submission were relatively minor, as I view the paper as a solid contribution to weight-space learning via probing. In the rebuttal, the authors addressed the majority of my concerns in a clear and satisfactory manner, and also introduced an additional theoretical result that modestly strengthens the paper. Overall, the discussion has reinforced my initial impression, and none of the points raised meaningfully change my evaluation. I therefore maintain my original positive assessment.

**Key Questions For Authors:**

1. **Clarification of weight selection**:
   The way you choose the weight matrix \( X \) is initially unclear and only clarified later in the paper. Introducing this earlier would improve readability.

2. **Symmetry vs. asymmetry discussion**:
   The discussion of “row/column asymmetry” is somewhat confusing in light of the earlier references to permutation symmetry in weight-space methods.
   It would help to explicitly distinguish between:
   - permutation symmetry of weights (a global invariance property), and
   - the row/column asymmetry introduced by the probing design.

   In particular, methods that operate on a single intermediate layer (including this one) are **not fully permutation-invariant**, and clarifying this would prevent confusion.

3. **Theoretical justification for \( X^T V \)**:
   While introducing the additional first-order probe \( X^T V \) is intuitive, it would strengthen the paper to provide a theoretical result demonstrating its benefit (analogous to Theorem 4.1).

4. **Comparison to ProbeX (X4) at the theoretical level**:
   The empirical comparison to ProbeX (X4) is important and well-motivated, since MVProbe effectively produces four probe matrices.
   A corresponding **theoretical comparison** (e.g., showing strictly greater expressivity than ProbeX (X4)) would significantly strengthen the theory section and might be more impactful than the current formulation of Theorem 4.1.

5. **Presentation of Theorem 4.2**:
   Theorem 4.2 appears somewhat abruptly. Adding a short paragraph explaining its goal and role in the overall argument would improve readability and flow.

6. **Related work**:
  It would be worth citing [1] in the final paragraph of the related work section. While it is not a direct baseline—due to differences in scalability—it explores similar ideas around incorporating higher-order information in probing, making it a relevant reference.



[1] Gelberg, Y., Eitan, Y., Navon, A., Shamsian, A., Putterman, T., Bronstein, M. M., & Maron, H. (2025). GradMetaNet: An Equivariant Architecture for Learning on Gradients. Advances in Neural Information Processing Systems (NeurIPS).

**Limitations:**

yes

**Strengths And Weaknesses:**

## Strengths

1. **Strong empirical evaluation**:
   The experiments are thorough and convincingly demonstrate improved performance over prior baselines, while keeping computational costs comparable.

2. **Simplicity and natural design**:
   The method is conceptually clean and builds naturally on prior probing approaches.

3. **Clarity of presentation**:
   The paper is well-written and easy to follow.

---

## Weaknesses

1. **Relatively weak theoretical section**:
   While not critical for the overall contribution, the theory could be strengthened to better support the proposed design choices (see suggestions below).

2. **Occasionally unclear motivation/intuition**:
   Some aspects of the method—particularly around symmetry and design choices—would benefit from clearer explanation and earlier clarification.

---

---

> ### Author Rebuttal · Authors · 2026-03-31
>
> For reproducibility (including rebuttal results), we share our anonymous code: https://anonymous.4open.science/r/mv1
>
> **Q1 (Weight selection clarity):**
> We agree that the weight selection procedure should be introduced earlier. Following ProbeX, we select the best-performing layer, and we will clarify this earlier in the paper in the revision.
>
> **Q2 (Symmetry vs. asymmetry):**
> We agree that the use of the term ''symmetry'' in two different contexts can be confusing to readers if not clearly separated. Permutation symmetry refers to a global invariance property of the full network's weight space under neuron reordering, which multi-layer methods must respect.
> Row/column asymmetry, on the other hand, is a property of the probing design: XU projects along columns only, ignoring the complementary row-space structure. As the reviewer correctly notes, single-layer probing (including ours) does not enforce full permutation invariance — it operates on a fixed layer where row and column dimensions have distinct semantic roles (output vs. input neurons). To prevent confusion, we will (i) explicitly distinguish these two notions early in Section 3 and (ii) consistently use the term "permutation equivariance" when referring to symmetry properties of weight-space representations.
>
> **Q3 (Theoretical justification for XᵀV):**
> We agree that clarifying the role of the column-side probe XᵀV would strengthen the presentation. Conceptually, XU and XᵀV capture complementary 1st-order views of the weight matrix. While XU summarizes how output neurons aggregate input directions, XᵀV summarizes how input coordinates interact with output neurons. We also provide a transpose-complement result: for any fixed row-side probe U with rank r < n, there exist distinct matrices X₁ ≠ X₂ such that X₁U = X₂U (identical row-side responses) but X₁ᵀV ≠ X₂ᵀV (distinguishable by column-side probing). The construction is: let w ∈ null(Uᵀ) and v ∈ ℝᵐ with Vᵀv ≠ 0, then X₂ = X₁ + vwᵀ gives identical XU but different XᵀV. This formalizes that XᵀV is not redundant with XU — it resolves complementary 1st-order blind spots that remain under row-only probing. While Thm 4.1 addresses the gap between 1st-order and 2nd-order, this result addresses the gap between row-side and column-side within the same order. Our ablations also provide empirical evidence consistent with this intuition: on SupViT, XU only (89.09) and XᵀV only (87.55) each miss information the other captures, and the 1st-order pair (89.88) outperforms either alone. We will include this result with proof in the revision.
>
> **Q4 (Comparison to ProbeX (×4) at theoretical level):**
> We agree that clarifying the theoretical relationship with ProbeX(×4) would strengthen the presentation. ProbeX(×4) increases the probe budget but remains within the same representation family as standard 1st-order probing. Concretely, it replaces U with a larger matrix Ũ ∈ ℝⁿˣ⁴ʳ, while the representation is still computed as XŨ. As a result, the same nullspace ambiguity discussed in Thm 4.1 still applies. If any rank(Ũ) < n, there exists w ≠ 0 such that wᵀŨ = 0, and therefore matrices that differ by a perturbation vwᵀ produce identical probe responses: (X+vwᵀ)Ũ = XŨ.
> This illustrates that increasing the number of same-side 1st-order probes enlarges the probe span but does not fundamentally change the representation class. In contrast, MVProbe improves not by using more probes, but by adding missing operator families: an opposite-side 1st-order view (XᵀV, see Q3) and interaction-aware 2nd-order views (XXᵀW, XᵀXZ). In the revision, we will strengthen this discussion by noting that this distinction is already reflected in the empirical results of Tab.1. It is further supported by our additional post-submission experiments on **Stable Diffusion LoRA** benchmarks, where the gains are especially large. Please see LtVP (Q1+W1) for the detailed experimental results.
>
> **Q5 (Presentation of Thm 4.2):**
> We agree that Thm 4.2 would benefit from a motivating paragraph. In the revision, we will add a brief introduction explaining that Thm 4.2 addresses a practical concern arising from the multi-order design: when combining 1st-order (O(σ²)) and 2nd-order (O(nσ⁴)) branches, the scale gap can cause 2nd-order responses to dominate the concatenated representation, degrading the contribution of 1st-order branches. This motivates per-sample standardization (Cor 4.3) as a principled solution.
>
> **Q6 (Missing citation — GradMetaNet):**
> Thank you for pointing this out. We will add it in the related work section.
>
> **W1+W2:**
> We addressed all suggested weaknesses above.

---

> > ### Author Rebuttal · Reviewer_HkVt · 2026-04-02
> >
> > I thank the authors for the thoughtful rebuttal. I have one follow-up question regarding Q3 and Q4.
> >
> >
> > While the provided constructions clearly demonstrate that MVProbe can distinguish weight matrices that standard first-order probing (including ProbeX (×4)) cannot, I am curious about the converse direction. Specifically, can you show that MVProbe is at least as expressive as ProbeX (×4), in the sense that it can distinguish any pair of weights that ProbeX (×4) can?
> > If this does not hold, it would suggest that the two approaches are theoretically incomparable, i.e., there may exist cases where ProbeX (×4) is preferable and others where MVProbe is. While the empirical results favor MVProbe, additional theoretical insight into this relationship would help clarify whether the improvement is strictly expressive or not.
> >
> > Overall, I appreciate the clarifications in the rebuttal and am leaning toward maintaining my positive score.

---

> > > ### Author Response · Authors · 2026-04-04
> > >
> > > Thank you for this insightful question.
> > >
> > > Our direct answer to the first question is **yes**. In the pairwise distinguishability sense, any weight pair distinguished by ProbeX(×4) can also be distinguished by MVProbe. If ProbeX(×4) distinguishes a pair (X₁, X₂), then there exist probe vectors U₁, …, U₄ such that (X₁U₁, …, X₁U₄) ≠ (X₂U₁, …, X₂U₄). Therefore, for at least one i, we must have X₁Uᵢ ≠ X₂Uᵢ. Since MVProbe contains the same first-order XU branch, choosing U = Uᵢ already separates the pair. Hence every pair distinguished by ProbeX(×4) is also distinguished by MVProbe.
> > >
> > > In fact, the number of probes assigned to each branch of MVProbe is only a hyperparameter. If we allocate the same total width to the XU branch, then ProbeX(×4) is simply the special case XŨ with Ũ = [U₁ U₂ U₃ U₄] inside the broader MVProbe family. In this sense, **ProbeX(×4) is contained in MVProbe**, and MVProbe is an extended model family that includes ProbeX(×4) as a subclass.
> > >
> > > Motivated by the reviewer’s insightful question, we also considered a comparison at the level of **fixed probe implementations** under the same total probe budget. Let us call the two variants **ProbeX(×4)** and **MVProbe**. ProbeX(×4) uses the entire probe budget in the first-order branch, whereas MVProbe distributes the same total budget across the branches XU, XᵀV, XXᵀW, and XᵀXZ.
> > >
> > > >### Definition (Uniform domination under fixed probes)
> > > >To formalize fixed-probe distinguishing power, fix a stacked first-order bank Ũ = [U₁ U₂ U₃ U₄] for ProbeX(×4), and fix one implementation of MVProbe with probes U, V, W, Z. Define H(X) = XŨ and G(X) = (XU, XᵀV, XXᵀW, XᵀXZ). We say that **MVProbe uniformly dominates ProbeX(×4)** if, for all weight pairs X₁, X₂,
> > > >
> > > >H(X₁) ≠ H(X₂) ⟹ G(X₁) ≠ G(X₂).
> > > >
> > > >Equivalently, uniform domination means G(X₁) = G(X₂) ⟹ H(X₁) = H(X₂).
> > >
> > > This differs from the pairwise argument above because the probes are fixed once and for all; we are comparing the distinguishing power of two fixed implementations rather than two probe families.
> > >
> > > >### Proposition (A sufficient condition for uniform domination)
> > > >If Col(Ũ) ⊆ Col(U), then MVProbe uniformly dominates ProbeX(×4).
> > > >
> > > >**Proof.** Assume Col(Ũ) ⊆ Col(U). Then there exists a matrix C such that Ũ = UC. Now suppose G(X₁) = G(X₂). Since XU is the first component of G, we have X₁U = X₂U. Therefore,
> > > >
> > > >H(X₁) = X₁Ũ = X₁UC = (X₁U)C = (X₂U)C = X₂UC = X₂Ũ = H(X₂).
> > > >
> > > >Thus G(X₁) = G(X₂) ⟹ H(X₁) = H(X₂). Taking the contrapositive, we obtain H(X₁) ≠ H(X₂) ⟹ G(X₁) ≠ G(X₂), which is exactly uniform domination.
> > >
> > > >### Corollary (A full-rank XU branch already covers any all-XU first-order bank)
> > > >If rank(U) = n, then MVProbe uniformly dominates ProbeX(×4).
> > >
> > > This observation clarifies the practical meaning of the sufficient condition. If the XU branch has not yet captured enough first-order directions, increasing its probe budget can still help. But once Col(U) becomes sufficiently rich allocating additional probes to the same branch no longer adds recoverable first-order information. At that point, further gains must come from complementary branches rather than from repeatedly enlarging the same first-order view.
> > >
> > > This is exactly why we believe the multi-view setting is the practically relevant one here. ProbeX is already an excellent first-order baseline. Therefore, once the number of XU probes becomes moderately large, it is unlikely that further performance gains will come simply from adding even more probes to the same branch. Instead, the remaining improvements are more likely to come from complementary views that expose row-side and second-order structure. This is also consistent with our experiments: increasing the number of first-order probes alone quickly saturates and can even reduce accuracy, while MVProbe—using the same total number of probes as ProbeX (×4), far fewer than ×10—substantially outperforms all scaled variants ([Link: baseline_saturates](https://anonymous.4open.science/r/mv1/a/f8.png)).
> > >
> > > We are very grateful for this insightful question. It helped us sharpen the distinction between simple pairwise inclusion and the stronger fixed-budget comparison between all-XU and evenly distributed multi-view designs. We will incorporate this discussion in the revision to provide a clearer theoretical guideline for when first-order expansion is sufficient and when complementary branches become necessary.

---

### Decision · Program_Chairs · 2026-04-30

**Decision:**

Accept (regular)

**Comment:**

The paper points out a real blind spot in how existing probing methods handle weight-space learning: standard first-order single layer probes  only see one side of the weight matrix. MVProbe fixes this with row/column projections plus Gram-based second-order views. The method is simple, cheap to run, and works well: consistent gains on Model Jungle across all architectures. All reviewers acknowledged their concerns were addressed. Final consensus is three weak accepts and one borderline-positive after discussion.

Required revision for camera-ready: Please add a direct comparison against invariant/equivariant weight-space methods: Zhang et al. (2023), Navon et al. (2023), Kofinas et al. (2024), and Lim et al. (2024)  on the MLP benchmarks where those methods were originally evaluated. Report accuracy, training time, inference time, and parameter counts. The current paper excludes these baselines, citing scalability constraints, which is fair for the large-scale setting, but a head-to-head on the smaller-scale regime that those methods can handle would clarify where lightweight probing suffices and where full equivariant processing is still needed